# Persuadable voters decided the 2022 midterm: Abortion rights and issues-based frameworks for studying election outcomes

**Claudia Kann**[ORCID][☯], **Daniel Ebanks**[ORCID][☯]*, **Jacob Morrier**, **R. Michael Alvarez**[ORCID]

Division of the Humanities and Social Sciences, California Institute of Technology, Pasadena, California, United States of America

☯ These authors contributed equally to this work.

* debanks@caltech.edu

**Data Availability Statement:** Upon publication, the data and code necessary to replicate the results reported in the paper will be archived in a public and community-supported repository.

## Abstract

Leading up to the 2022 Congressional midterm elections, all predictions pointed to a Republican wave, given factors such as the incumbent president's low approval rate and a struggling national economy. Accordingly, the underwhelming performance of the Republican Party surprised many, resulting in an election that became known as the "asterisk election" due to its unusual and seemingly unpredictable outcome. This study delves into the specifics of the 2022 midterms, exploring factors that may have influenced the results beyond those traditionally considered by political scientists. Our analysis particularly seeks to understand whether a sudden shift in the public salience of specific issues could have influenced voters' preferences, leading them to consider factors they might not have otherwise. To achieve this, we analyzed data from a nationally representative sample of registered voters surveyed immediately after the midterm elections. Our findings reveal that the issue of abortion played a pivotal role during this election. The prominence of abortion was not predestined, as evidenced by a comparative analysis with data from a survey conducted after the 2020 presidential election. Indeed, it seems that the decision by the Supreme Court to overturn *Roe v. Wade* in June 2022 significantly increased the salience of abortion. This unexpected policy shock had a significant impact on the behavior of voters in the 2022 midterm elections.

## 1. Introduction

A recurring trend in contemporary American politics is the tendency for the incumbent president's party to underperform during Congressional midterm elections [1]. Political scientists have proposed several explanations for this phenomenon, in particular that it arises because voters perceive midterm elections as a referendum on the economic performance of the president and their party [2, 3]. In line with this expected tendency, the 2018 midterm election adhered to a referendum-like pattern, as evidenced by recent research [4]. During this election, the Republicans suffered a significant setback, losing 40 seats in the House of Representatives, though they gained two seats in the Senate.

**Funding:** The John Randolph Haynes and Dora Haynes Foundation (https://haynesfoundation.org/) and the Resnick Sustainability Institute (https://resnick.caltech.edu) provided funding to RMA to support the collection of the survey data used in this research. The funders had no role in study design, data collection and analysis, decision to publish, or preparation of the manuscript.

**Competing interests:** The authors have declared that no competing interests exist.

Prior to the November 2022 midterm elections, pre-election analyses, which took into account retrospective and contextual factors such as the president's approval rating, the state of the national economy, and redistricting, indicated that the Democratic Party was likely to face seat losses in both the House and Senate. Undoubtedly, the forecasts for the 2022 midterm elections exhibited significant variations in the projected extent of Democratic seat losses [5]. For instance, one analysis on August 30, 2022 predicted the Democrats would lose 30 House and three Senate seats [6]. Conversely, Jacobson noted that a conventional predictive model, based on presidential popularity and economic factors, would have expected a Democratic House seat loss of around 45 seats. However, he also acknowledged that due to partisan loyalties and issues like abortion, the actual Democratic seat loss could be notably lower than that [7]. Finally, Larry Sabato's "Sabato's Crystal Ball" forecasted that the Republicans would gain one seat in the Senate and 24 seats in the House [8]. Indeed, amid widely reported and significant levels of inflation, the incumbent president, Joe Biden, was relatively unpopular [9, 10]. Despite these challenging circumstances, the Democratic Party experienced a relatively minor setback, losing only nine House seats, while concurrently strengthening their Senate majority to 51 seats. This particular outcome, depicted in Fig 1, marked a historically mild loss, diverging from the forecasts of conventional models typically employed to predict midterm election results.

Additional evidence suggests that the 2022 midterm elections were an anomaly. Remarkably, for the first time since 1934, the party in power at the White House maintained control of every single state legislature, even securing full control of the state government in a crucial swing state, Michigan. Furthermore, the Democrats successfully expanded their legislative majorities in Nevada and California. Notable Democratic victories in key gubernatorial races across multiple swing states, including Arizona, Michigan, Pennsylvania, and Wisconsin, added to their impressive performance. It is important to acknowledge that the Republicans also experienced successes, making gains in states like Florida and Texas. Additionally, New York swung in favor of the right-wing, granting the Republican Party its entire margin of control in the House of Representatives. Despite these Republican achievements, the Democrats' performance during the midterm defied expectations in a positive manner.

This paper delves into the under-performance of the Republican Party in the 2022 midterm elections. Despite a context where presidential approval and the economy appeared to be in

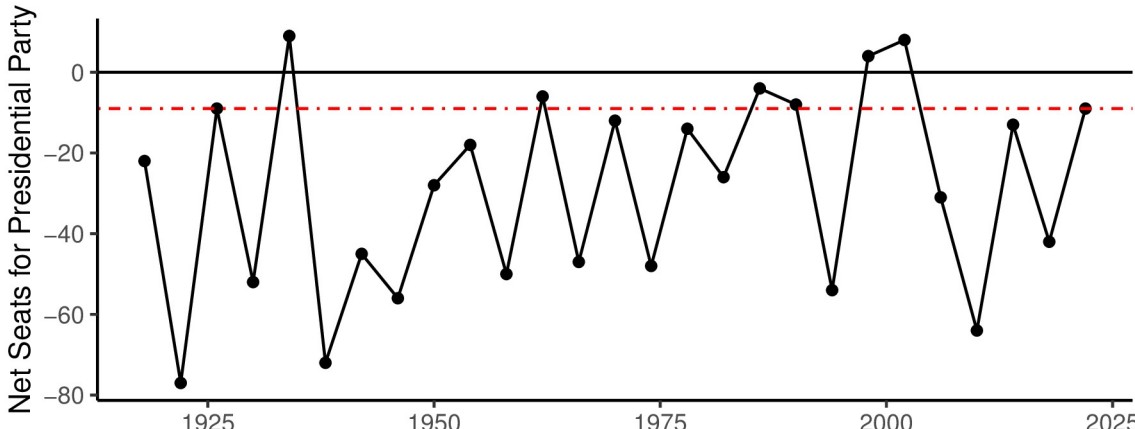

**Fig 1. House seat change for presidential parties during midterm years.** The three times where seats were gained (1934, 1998, and 2002) can be seen. In addition, it is clear the 9 seats lost in 2022, represented by the red dashed line, were historically mild losses. Source: History, Art & Archives, U.S. House of Representatives, (https://history.house.gov/Institution/Party-Divisions/Party-Divisions/).

their favor, the anticipated substantial gains for Republicans in both the House and the Senate did not materialize. We contend that factors beyond the control of President Biden and Congress are responsible for explaining this seemingly anomalous outcome. Specifically, certain external factors, such as the U.S. Supreme Court's *Dobbs v. Jackson Women's Health Organization* decision, exerted a significant influence on segments of the midterm electorate. These voters cast their ballots based on issue-oriented considerations, prioritizing policy matters such as abortion over their approval of President Biden or the state of the national economy.

Our study makes a significant contribution by acknowledging the importance, particularly during times of heightened polarization, of investigating the determinants that influence voting choices among persuadable voters—those who do not strongly align with any political party. To achieve this objective, our paper utilizes individual-level survey data to conduct a comprehensive investigation into the factors associated with voting decisions in the 2022 midterm election. Given the prevailing political climate, where Democratic and Republican identifiers tend to support candidates from their respective parties, the use of individual-level survey data becomes crucial for conducting an in-depth analysis of this unusual election.

Our research focuses on studying the voting decisions of three distinct groups: Democratic, Republican, and notably, Independent voters. By examining a nationally representative survey of 2,109 registered voters, we provide compelling evidence that Democratic voters predominantly voted along party lines. However, our findings reveal that the issue of abortion played a decisive role in persuading enough Independents and Republicans in key districts, leading to a draw in the House.

The subsequent sections of the paper are structured as follows. We begin by introducing the data collection process and survey methodology utilized in this study. Following that, we present crucial descriptive statistics, which form the foundation for our subsequent modeling decisions and regression strategy. Next, we unveil the key findings derived from our analysis, providing a rationale for our chosen model specification and drawing comparisons to analogous results from the 2020 election to strengthen our conclusions. Lastly, we conduct a thorough examination of the results and discuss their implications for election studies.

## 2. Midterm elections and American politics

Early research on midterm elections in the United States focused on the so-called "surge-and-decline" theory [11]. Building on the same concept of partisanship that was articulated in the seminal book *The American Voter*, the basic argument was that during presidential election years the winning president's party would gain seats due to the short-term salience of partisanship. But in the midterm elections, the salience of party would recede and thus the out-party would gain seats in midterm, with midterm elections reflecting the partisan equilibrium in the nation. As such, the seat distribution following the midterm elections should be normally distributed around this stable equilibrium.

However, while in general there is a swing in House seats away from the president's party, there is historically a great deal of volatility in the magnitude of this midterm swing [12]. There is much more volatility than is explained by the return to a normal vote [13]. Clearly, for many midterm elections, there is another component to the seat shift away from the president's party which accounts for the magnitude of loss. Compelled by this volatility, researchers started to focus on the association between other retrospective performance factors and election performance.

The first model of this sort was the midterm-as-a-referendum model, which linked the magnitude of the downward swing with the president's approval and the state of the national economy before the midterm elections [2, 3]. Voters were seen to punish the party of the

President for their view of him as well as for how, what they viewed as his actions, were affecting the country. This model has seen general support from historical data [2–4, 14] and is often the basis for many midterm elections predictions.

The midterm-as-a-referendum model generally focuses on two factors: the president's approval rating, and the state of the national economy. In situations where the president's approval rating is low and where economic performance is poor, the president's party should lose a large number of Congressional seats. When these factors are ambiguous, for example when a president's approval is low but the economy is performing well (like in the context of the 2018 midterm election), election losses for the president's party may be more muted.

Building on this retrospective approach, various scholars have proposed a balancing model of voter behavior to explain the mechanisms underlying the midterm backlash [15–17]. This model synthesizes both retrospective and prospective elements. Voters retrospectively consider the president's legislative agenda and deem it extreme. They then reward the opposition and punish the president's party to constrain her future legislative options. Here, the median voter exploits the checks and balances of a presidential system to forcibly moderate the president by handing control of the legislature to the opposition party. If the president and legislature wish to pass laws, they will need to find consensus.

Evidence used to test the balancing model often relies on aggregated measures of electoral outcomes [15, 18], even in comparative contexts [19]. In aggregate, it is not clear that the 2022 midterm elections can easily be explained by balancing models. The model's overall prediction —that voters would likely want to balance the second two years of President Biden's term by giving Republicans strong majorities in the House and Senate—did not occur. Other researchers have tested the balancing model using different approaches and the empirical evidence does not tend to provide support for the model [20–22]. Finally, balancing theories also imply that voters engage in a complicated cognitive process—involving both retrospective and prospective elements. These assumption seems at odds with empirical research that shows that voters are generally poorly informed and unsophisticated [23, 24].

Existing models of U.S. midterm elections seem to not explain the outcome of the 2022 midterm, which means we must turn to other models of voter decision making. If the 2022 midterm elections were not decided by retrospective evaluations of President Biden's or the Democratic Party's performance, nor the state of the national economy, nor by sophisticated strategizing about balancing the power of the two parties across the three branches of government, what other theories of voter decision making might help explain this election?

The other factors often used to explain voting behavior in American federal elections are partisanship and issues. Partisanship is a powerful factor in American politics and has long been shown to be a key decision variable for voters [25]. However, in recent elections in the United States for many in the electorate their party affiliation has become synonymous with their voting decisions [26]. Virtually all Democratic identifiers vote for Democratic candidates, while virtually all Republican identifiers voter for Republican candidates.

Thus, partisanship is a key part of our story for the 2022 midterm elections: since partisan identifiers vote for their party's candidates, we need to study those who do not identify with a party, those who are Independents [27]. In the context of today's highly polarized political environment in the United States, where party identification is synonymous with voting decisions, the political independents are the potentially persuadable voters [28].

This is also where political issues enter the story. Political independents lack the pull of partisanship, and if retrospective factors are not pushing their voting decisions, then perhaps highly salient political issues will dictate how they vote in midterm elections. Research has shown that uncertain voters may cast their ballots based on issue information [29] and here we note that in 2022 there were highly social and policy salient issues like abortion, gun policy,

COVID-19, foreign policy, racial and ethnic inequalities. However, the past survey-based research has advanced a number of different methods for evaluating the role that issues may play in a certain election. One approach has been to include issue information, usually based on voter placement of themselves and candidates on spatial issues; those studies then use different statistical models to produce estimates of the importance of each spatial issue in the model (e.g, [30, 31]). A second, and more recent approach, uses a choice-based methodology to estimate in a causal inference framework the importance of each issue in a particular electoral context (e.g., [32]). Finally, many studies have used different direct survey questions, asking voters to rate the importance of different issues themselves, then using those ratings in models of voting behavior (e.g., [33, 34]). We take this third approach in our research, and we discuss the details regarding our survey methodology and modeling approaches below and in the paper's online S1 Appendix.

But in this paper we argue that the abortion issue that was the focal point of the 2022 midterm elections. In the United States, the 1973 U.S. Supreme Court decision in *Roe v. Wade* established abortion as a constitutional right for people with uteruses. This right was largely confirmed by subsequent Supreme Court decisions like *Planned Parenthood v. Casey* in 1992. During this period abortion became a divisive issue, part of the partisan landscape of American politics. [35, 36] For decades, while partisan voters had distinct positions on abortion, it did not seem that elected officials had much say in the matter as *Roe v. Wade* generally established the constitutional right to abortion. But in the summer of 2022, as congressional campaigns started to take shape, the Supreme Court shocked the political world by handing down the *Dobbs v. Jackson Women's Health Organization* decision, which held that the Constitution does not provide a right to abortion. This was a true shock to the American political system, and suddenly the issue of abortion again became salient as legislatures at the state and federal levels became the focus of debate about the future of abortion policy in the United States.

The referendum model has generally done well explaining past midterm elections outcomes. However, some initial support for an issue-based model come from observational results of past elections. In all but three midterm elections since 1916, the president's party has suffered a net loss of seats in the House of Representatives, as seen in Fig 1. In each of the three anomalous cases where the president's party gained seats during the midterm, there were clear external factors contributing to the White House's party success (e.g. the Great Depression in 1934, the Clinton impeachment in 1998, and the 9/11 terrorist attacks in 2002). Thus, while it seems that the recent performance of the president's party and the overall state of the national economy help determine the makeup of Congress after a midterm election, other issues may arise that can lead to anomalous outcomes.

## 3. Data and methodology

### 3.1 Data collection process

In this study, we primarily relied on data from a nationally representative online survey conducted in the aftermath of November 2022. Our research team designed the survey as part of a broader project aimed at understanding the opinions and political behavior of the American electorate. YouGov carried out the survey, and prior to its implementation, the survey design underwent review and approval by Caltech's Institutional Review Board (IRB). Informed consent was waived by the IRB to facilitate data collection. To protect the privacy and confidentiality of the participants, YouGov provided us with a fully anonymized data set.

The survey was conducted in the days immediately following the 2022 Congressional midterm elections, specifically from November 9 to November 19, 2022. Our sample consisted of 2,109 U.S. registered voters, carefully selected by YouGov from their opt-in survey subject

panel. The survey's margin of error is approximately 2.3%. For our analyses, we employed sample weights provided by YouGov. These weights were computed based on data from the American Community Survey, incorporating information on gender, age, race, education, and the 2020 Presidential vote. They have a mean of 1.0, a standard deviation of 0.4, and range from 0.1 to 4.2. All the estimates presented in this paper have been weighted to ensure an accurate representation of the population.

Furthermore, we incorporated data from a separate survey carried out after the presidential elections in November 2020. This survey also included a nationally representative sample of American registered voters and was conducted online by YouGov. The 2020 survey comprised comparable questions to those used in our November 2022 survey, enabling us to draw comparisons between the data sets and election periods. Similar to the 2022 survey, the design of the 2020 survey obtained approval from Caltech's IRB, and informed consent was waived.

The 2020 survey was conducted from November 4 to November 10, 2020, and YouGov recruited subjects from both their opt-in survey subject panel and an external partner to ensure diversity in the sample. The total sample size for the 2020 survey was 5,051, with an estimated margin of error of 2.0%. To ensure representativeness, the survey data were weighted using various factors, including gender, age, race, education, U.S. Census region, state of residence, and the 2020 Presidential vote. The weights ranged from 0.1 to 5.973, with a mean of 1 and a standard deviation of 1.

Our research team works with with YouGov to conduct surveys in a way that addresses potential sampling bias, selection bias, and non-response issues. Adhering to the standards set by the American Association for Public Opinion Research (AAPOR) for survey collection, we took meticulous care to ensure our survey captured a nationally representative sample of engaged voters. To achieve this goal, YouGov provided us with fully anonymized datasets that use scientific weighting procedures. The weights (which we using in our analyses reported below) incorporate variables such as education, age, race, and gender. The weighting process took into account respondents' voting choices to minimize any influence of partisan-response bias. These efforts were aimed at obtaining high-quality responses that authentically reflected the genuine opinions of the participants. By implementing these rigorous strategies, we sought to address potential biases and produce reliable insights.

Nevertheless, we must acknowledge that no survey is entirely immune to biases and limitations. For instance, we recognize that our survey may have attracted politically engaged and well-informed individuals, given its time-consuming nature, despite offering compensation to respondents. While it was challenging to entirely eliminate this bias, we took measures to address it by including education in our analyses, which is closely linked to political engagement and awareness, thereby partially mitigating this concern. However one limitation of our methodology is that these steps may not necessarily alleviate potential biases due to issues like the opt-in, fully-online nature of this survey and non-response issues.

Also, like any survey-based analysis, we are relying on the respondents to provide accurate and truthful answers to our questions. They are self-reporting important information for our analysis, like who they voted for, their opinions on important issues, and their evaluations of the economy. Our research group has extensive experience with survey questionnaire design, and we used questions that we have used in the past and which are similar to those generally used in academic research. We extensively pre-tested our survey questionnaire before implementation and undertook various data quality analyses prior to our use of the data in this paper, procedures which help to insure the quality of our survey data.

Another potential issue is the impact of media narratives immediately after the election on voters' responses. The majority of respondents completed their surveys in the days immediately following the election, before media narratives had solidified. Additionally, the

declaration of winners in both the 2020 and 2022 elections occurred approximately a week after the election, primarily due to the counting of mail-in votes. This timing should have minimized biases introduced by voters falsely claiming to have voted for the winner before the official declaration had taken place. Besides, since congressional elections typically involve two-candidate races, strategic voting is not a prominent concern in our study.

Furthermore, our analysis does not include certain issues that were not part of the survey. Among these is election denialism and the broader threats to American democracy that emerged after the 2020 election. Although election denialism might have influenced voter decisions during the 2022 midterms, it is not expected to have a significant correlation with social issues like abortion, and therefore, it is unlikely to introduce any bias into our findings. It is worth mentioning that recent unpublished research conducted by [37] indicates that election denialism had a minimal impact on midterm voters.

Finally, we motivated our paper by discussing expectations from academic and media pundits about the potential outcomes of the 2022 midterm elections, which are framed in terms of seat gains and losses. Our work uses individual-level survey data to study the factors that are associated with voting decisions in the midterm election; our data does not give us the ability to aggregate to the congressional district level, certainly not for all of the congressional races nationwide in 2022. While we suggest that other researchers study midterm elections using survey data like ours, we also suggest that future research considers the important questions of studying seat shifts in the 2022 election.

## 3.2 Variables of interest

In our analysis, we integrated a diverse range of variables derived from the survey participants' responses. To promote transparency and provide a point of reference, we have included the complete wording of all survey questions in the S1 Appendix.

The main dependent variable in our analysis is the Congressional midterm vote, which reflects the voters' preferences in the election. To gather this information, we utilized a generic ballot question, asking respondents: "In the November 2022 election for U.S. Congress in your district, which candidate did you vote for?" Participants were given the opportunity to indicate whether they voted for the Democratic candidate, the Republican candidate, neither, were unsure, or did not vote at all.

In our survey, we incorporated two questions specifically crafted to assess respondents' perceptions regarding their personal finances and the state of the national economy:

(i) **Personal Financial Situation**: We are interested in how people are getting along financially these days. Would you say that you and your family living here are better off or worse off financially than you were a year ago?

(ii) **National Economic Situation**: Now thinking about the economy. Would you say that over the past year the nation's economy has gotten better, stayed the same, or gotten worse?

These questions are of utmost importance in testing the midterm-as-a-referendum model and are similar to those used in previous research [38, 39].

Previous research has extensively investigated two main approaches to assess the importance of issues in elections and their influence on voters' decision-making. A persistent debate revolves around the merits of using self-reported measures versus choice-based measures of issue importance [29, 31–34]. In our surveys, we integrated numerous self-reported measures for issue importance, encompassing a wide range of relevant topics.

We obtained one measure from a question that inquired about participants' perceptions regarding which party, Democratic or Republican, they believed to be more capable of

effectively addressing various issues. These issues encompassed a broad range of topics, such as terrorism, climate change, abortion, law enforcement and criminal justice, the COVID-19 pandemic, the federal budget deficit, economic growth, healthcare, foreign policy, and inflation. During the analysis of this measure, we encountered notable collinearity issues with other partisan indicators present in the survey. It quickly became evident that respondents consistently tended to rate their own party favorably in terms of competence when dealing with policy issues.

To circumvent these collinearity issues, we utilized responses from a different question that asked participants to assess the absolute importance of various policy issues. The specific question was: "How important, if at all, were each of the following issues for you as you thought about whom you would vote for in the congressional election in your area in November 2022?" The list of issues included immigration, abortion, foreign policy, economic inequality, the COVID-19 outbreak, violent crime, health care, the economy, racial and ethnic inequality, climate change, inflation, gun policy, and Supreme Court appointments. For each policy issue, respondents were given the following response options: "very important," "somewhat important," "not too important," and "not important at all." To simplify the analysis, we converted these choices into a binary scale. Specifically, we assigned a value of 1 to issues labeled as somewhat or very important, and a value of 0 to those considered not too important or not important at all. Any unanswered questions were treated as missing data and were excluded from the analysis. In contrast to assessments of parties' competence in addressing policy matters, this binary measure of issue importance demonstrates a weaker correlation with party identification. Nonetheless, it exhibits significant variation, allowing us to draw meaningful conclusions about the specific issues that influenced voters' decisions.

We integrated responses from a three-point partisanship question into our analysis. This question asked participants about their political affiliation, providing options for them to identify as Republican, Democrat, or Independent. It is important to highlight that this question was specifically designed to focus on self-identification rather than party registration. We adopted this approach to prevent potential complexities related to party registration, particularly concerning primary rules, and to gain deeper insights into the respondents' partisan loyalty. We note that there are several valid approaches to classify how a voter self-identifies their party affiliation. For example, the standard approach to party identification is to classify self-identified leaners as Independents. We follow this approach in the main results reported in this paper. As a robustness check, we re-run the main analysis except we re-classify self-identified leaners as being affiliated with their respective parties. We find the results are largely consistent with our main analyses. Figs C-F in S1 Appendix all show results under using a coding approach where leaners are excluded from independents and included with their parties do not materially change the main results. Differences from the main results are further explored in the S1 Appendix.

Lastly, we considered several demographic factors, including gender, educational attainment, region, race and ethnicity, religious affiliation, and age. These factors were incorporated to account for other potential influences on voting decisions.

## 3.3 Descriptive statistics

Before delving into our analysis, we conduct an exploratory examination of the distribution of survey responses. This process aims to identify any significant patterns or trends that may have relevance to our investigation.

Our primary focus is to assess partisan polarization, specifically examining whether individuals voted for the party they identified with. The results of this investigation are illustrated in

Fig 2. Remarkably, individuals who identified as Democrats displayed a notably strong inclination to vote for the Democratic candidate. Conversely, among Republicans, this association was less pronounced, and Independents exhibited a split in their voting patterns, selecting candidates from both parties. This intriguing observation leads us to concentrate our study on Independent and Republican voters, as their voting behavior appears to have influenced the outcome away from the Republican Party.

Given the midterm-as-a-referendum model's proposition that vote choice is influenced by perceptions of the economy and personal finances, we now delve into an examination of the distribution of vote choice based on these perspectives, in addition to party identification. The corresponding findings are depicted in Fig 3. Notably, individuals identifying as Democrats

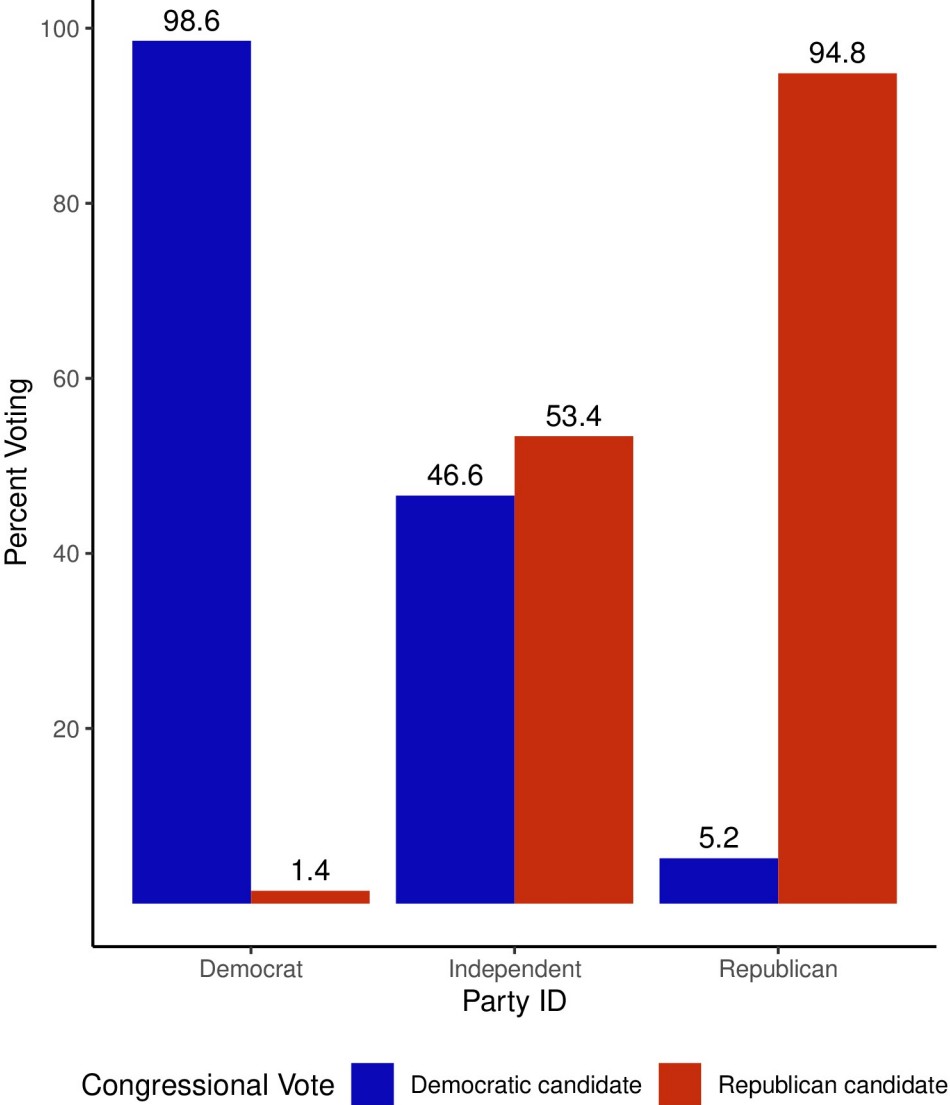

**Fig 2. This figure shows the weighted percentage of members of each party who voted for congressional candidates of each party.** From this figure it is clear that party cohesion was strong—most people voted with their party. This supports the idea that most of the interesting variation will come from those who identify as Independents. Independents as a group make up about half as many individuals as Democrats or Republicans, indicating that they could be the swing vote in the election. Democrats were significantly more loyal to their party than Republicans.

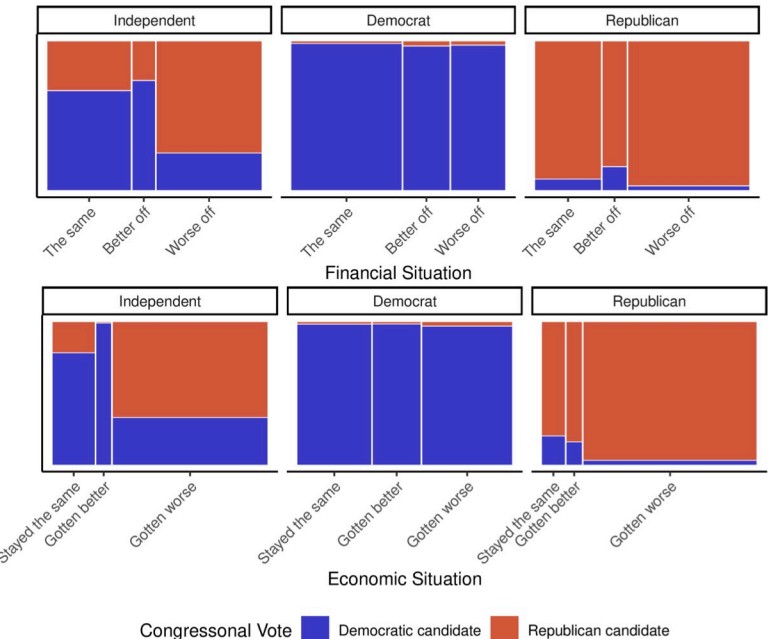

**Fig 3. Vote choice by party identification and views of the national economy and personal finances as compared to the previous year.** The width of the bars represent what proportion of individuals (weighted) fell into each grouping. This helps to see how the answers are related to vote choice but also how party identification relates to the response.

consistently and strongly lean towards voting for the Democratic candidate, regardless of their views on the financial and economic situation. Democrats generally express a positive outlook on both fronts, with less than half of the respondents believing in any deterioration. On the other hand, Independents exhibit a different voting pattern. When they perceive a decline in the economic or financial situation, they are more likely to vote for the Republican candidate. A significant proportion of Independents feel that the economy has worsened, and almost half hold a similar view regarding their personal financial situation. Interestingly, Republicans show a slight inclination to vote for the Democratic candidate if they believe the economic or financial situation has improved or remained stable. However, it is essential to acknowledge that the number of individuals holding this view is relatively small, particularly concerning the economy. While these findings align with certain aspects of the midterm-as-a-referendum model, there is a noticeable leaning towards the Democratic Party that the model fails to fully account for.

The analysis of voter behavior becomes more complex when considering the divisions on issues within the electorate, as depicted in Fig 4. When examining individuals affiliated with the Democratic or Republican parties, it appears that specific issues did not consistently influence their voting choices, except for minor exceptions observed in the case of the economy, inflation, and violent crime. For example, when Republicans perceived these issues as unimportant, they were more inclined to vote for the Democratic candidate. However, it is crucial to note that these instances were relatively rare in the overall population. In contrast, among Independents, a more pronounced differentiation emerged based on specific topics. Independents who considered issues like abortion, climate change, COVID-19, economic inequality, gun policy, healthcare, or racial and ethnic inequality as important had a significantly higher likelihood of voting for the Democratic candidate. Conversely, Independents who prioritized

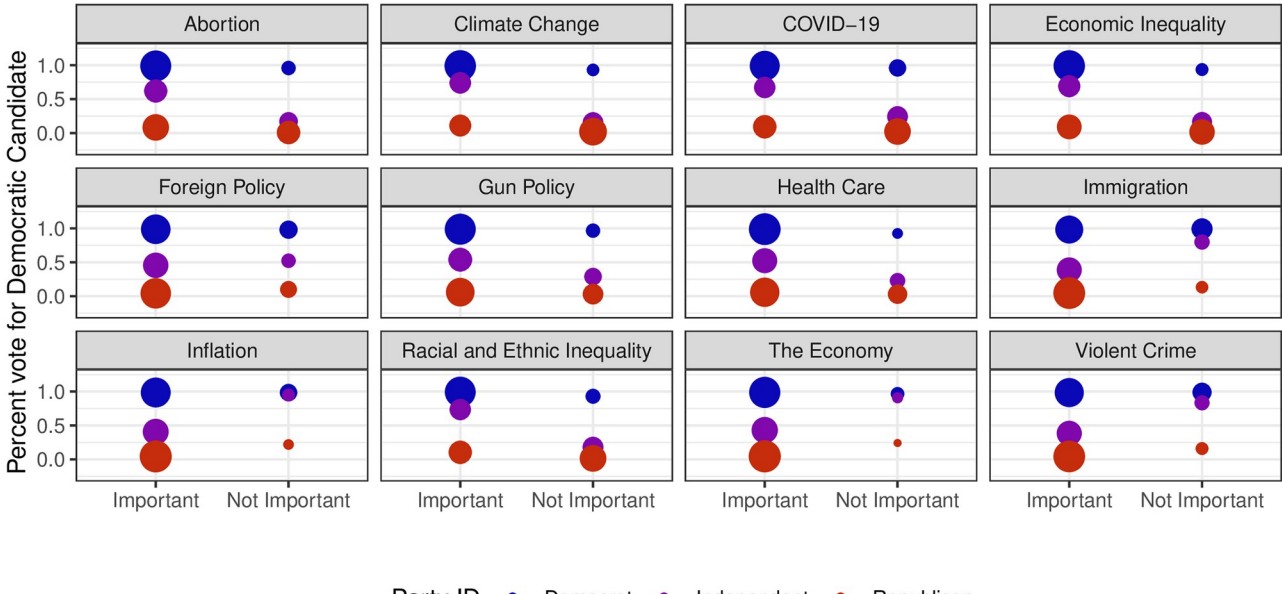

**Fig 4. For issues, how different party identifiers voted based on whether they thought it was important or not.** The color represents partisan identification while the size is the weighted number of individuals who fit the category. It is clear that most partisans stuck to their party, regardless of their views on issues. Independents were swayed by the issues they viewed as important.

immigration, inflation, the economy, or violent crime were more prone to vote for the Republican candidate. These findings emphasize the substantial impact of issues on shaping the voting choices of Independents. Consequently, exploring the significance of various issues could be a promising area of research, particularly considering evidence that challenges the validity of a midterm-as-a-referendum model in explaining final election results.

## 3.4 Multivariate analysis strategy

The primary model we employ to assess the predictive relationship between demographics, opinions, and vote choice is a comprehensive logistic regression model. In the initial stage of our analysis, we include a set of demographic controls, comprising variables such as race, gender, age, region, and educational attainment. Additionally, we integrate variables that capture individuals' perceptions of which party is more adept at handling various issues, along with variables representing the importance they assign to specific issues.

$$Pr_{(\text{Vote}=Rep|\text{Demo, Issues, PID3})} = \frac{\exp(\beta_0 + \beta_1\text{Demographics} + \beta_2\text{PID3} + \beta_3\text{Issues})}{1 + \exp(\beta_0 + \beta_1\text{Demographics} + \beta_2\text{PID3} + \beta_3\text{Issues})} \qquad (1)$$

During our analysis, we encountered a significant issue of high collinearity between the questions related to party superiority on issues and party identification. To ensure the accuracy of our findings, we made the decision to exclude the questions concerning party superiority on issues from analysis. Furthermore, to streamline the model and enhance its clarity, we eliminated questions that were not statistically significant. For a more detailed description of this process, please refer to Appendices A.4 to A.5 in S1 Appendix. This model is commonly known as the "pooled" model, which we outline in Eq 1. Note that $\beta_1$ and $\beta_3$ are vectors of logit coefficients representing the coefficients on the included demographic and issue-importance

variables.

$$Pr_{(\text{Vote}=Rep|\text{Demo, Issues, PID3=p})} = \frac{\exp(\beta_0 + \beta_1 \text{Demographics} + \beta_2 \text{Issues})}{1 + \exp(\beta_0 + \beta_1 \text{Demographics} + \beta_2 \text{Issues})} \quad (2)$$

Furthermore, we employed a model referred to as the "party-based" model, where we estimate the same model as in the pooled model shown in Eq 1 separately holding Party ID fixed. Eq 2 outlines this logit specification. Note that $\beta_1$ and $\beta_2$ are vectors of logit coefficients representing the coefficients on the included demographic and issue-importance variables. This approach involves analyzing individuals of each party identification separately. The main objective of this model is to concentrate on Independent voters, who seemed to play a crucial role in the election, as well as Republican voters, who exhibited more significant deviations from the party line when compared to Democratic voters. The advantage of using the party-based model is that it enables us to explore distinct effects of each covariate on individuals belonging to each group.

The decision to adopt a party-based model is reinforced by a series of Wald tests, which are thoroughly described in A.5 in S1 Appendix, and by compelling evidence presented in Fig 2. This evidence underscores the highly polarized nature of the contemporary American electorate. According to the survey data, nearly all Democratic partisan identifiers reported voting for Democratic candidates, while almost all Republican partisan identifiers in our sample voted for Republican candidates. This finding suggests that strong partisans were not meaningfully persuadable during the 2022 midterm election. Consequently, our primary focus is on Independents depicted in the middle columns of Fig 2. Notably, their support for House candidates was nearly evenly divided between Democratic and Republican candidates, indicating the significant influence Independents had on the outcome of the 2022 midterm elections.

Our study relies solely on observational data, and our primary objective is to assess the predictive strength of the variables under investigation. Therefore, we refrain from making any causal claims in this study. Instead, we employ robust statistical tests to gather evidence that either supports or challenges existing theories regarding midterm electoral behavior. Specifically, we conduct a meticulous analysis of the average marginal effect of responses to various questions on vote choice. This approach enables us to systematically explore the relationships between these variables and voting behavior. The statistical findings derived from these analyses serve as the foundation of our study's conclusions and inferences.

If the November 2022 elections had followed the "midterm-as-a-referendum" model, we would expect to observe a significant average marginal effect of voting Republican when respondents perceived the economy and/or their individual financial situation to have worsened. Conversely, we would anticipate a negative marginal effect when respondents expressed views of improvement. Moreover, after accounting for demographics and party preference, we would not expect any significant effect of issue importance on vote choice.

Finally, if the election was driven by specific issues, we would anticipate identifying certain issues with statistically significant negative average marginal effects. These issues would likely influence voters to support the Democratic candidate.

We test these expectations in the next section.

## 4. Results

According to two widely accepted theoretical models—the midterm-as-a-referendum model and the balancing theory model—the Republicans were expected to achieve substantial gains in this election. Indeed, the conditions required for both models seemed to be met. The midterm-as-a-referendum model relies on an unpopular sitting president, middling to poor

economic perceptions, and a strong campaign by the opposition party on presidential performance—all of which were evident during the election. Similarly, the balancing theory model looks at the Democrats' control of both the House and Senate, along with the presidency, and the passage of significant spending legislation in the previous session, which should have favored the Republicans in this election.

Despite meeting the conditions outlined by both theoretical models, the election results defied expectations. The Republican Party managed to secure 221 seats in the House, representing a modest increase of nine seats from the previous election. However, their performance in the Senate went in the opposite direction, experiencing a decline of one seat, with only 49 seats obtained. As a consequence, the Republicans fell short of attaining the significant majorities in both the House and Senate that had been anticipated, and their expected dominance in state-level offices, particularly in swing states, did not come to fruition.

To gain a deeper understanding of the factors contributing to this unforeseen result, we conduct additional testing on the two prevailing theoretical models, with a specific emphasis on the U.S. House. Our decision to focus on the House is driven by the fact that all 435 districts hold elections, enabling us to study the entire sample of registered voters. Moreover, House races are less susceptible to the influence of candidate recognition compared to Senate or Governorship races, making them a more suitable arena for observing the effects of the theoretical models on midterm elections.

We initiate our investigation by closely examining the midterm-as-a-referendum model. Our analysis indicates that there is minimal support for this model at the individual level, particularly when evaluating self-reported assessments of the in-party's economic performance. Testing the balancing theory model proves to be a more complex endeavor. Any effort to elicit voters' strategic considerations is likely to be heavily correlated with party identification and voter ideology, making it difficult to draw definitive conclusions. Also, it is important to note that the results of the election show the Democrats secured control of the House, Senate, and Presidency, suggesting that, in this specific case, the balancing model did not heavily influence the election. Consequently, the balancing theory model does not emerge as a strong explanatory candidate for the 2022 midterm elections, nor for midterm elections in general.

Considering the limited empirical support for the midterm-as-a-referendum model and the challenges in testing the balancing theory, we propose an alternative model to explain the outcomes of the 2022 midterm elections. Our research highlights the influential role of issues in shaping voter decisions, particularly among persuadable individuals, encompassing both political and weak partisans. Surprisingly, these persuadable voters showed unexpected support for Democrats, primarily driven by their alignment with specific issues they considered significant. Notably, abortion emerged as a major indicator of Democratic support, while crime and inflation were key factors correlated with Republican support. Finally, to gauge the novelty of the effect of abortion, violent crime, and immigration on voter choices, we conducted a comparison of the 2022 midterm election results with a benchmark from the 2020 elections.

## 4.1 Testing the referendum model

To begin, we investigate the seminal midterm-as-a-referendum model. After analyzing the aggregate results, it becomes evident that the model does not offer a satisfactory explanation for the election's outcome. This is apparent from the Republican Party's relatively underwhelming performance, despite favorable factors such as high inflation and a low approval rating for the President. However, in pursuit of deeper insights, we proceed with an investigation at the individual level to validate this observation.

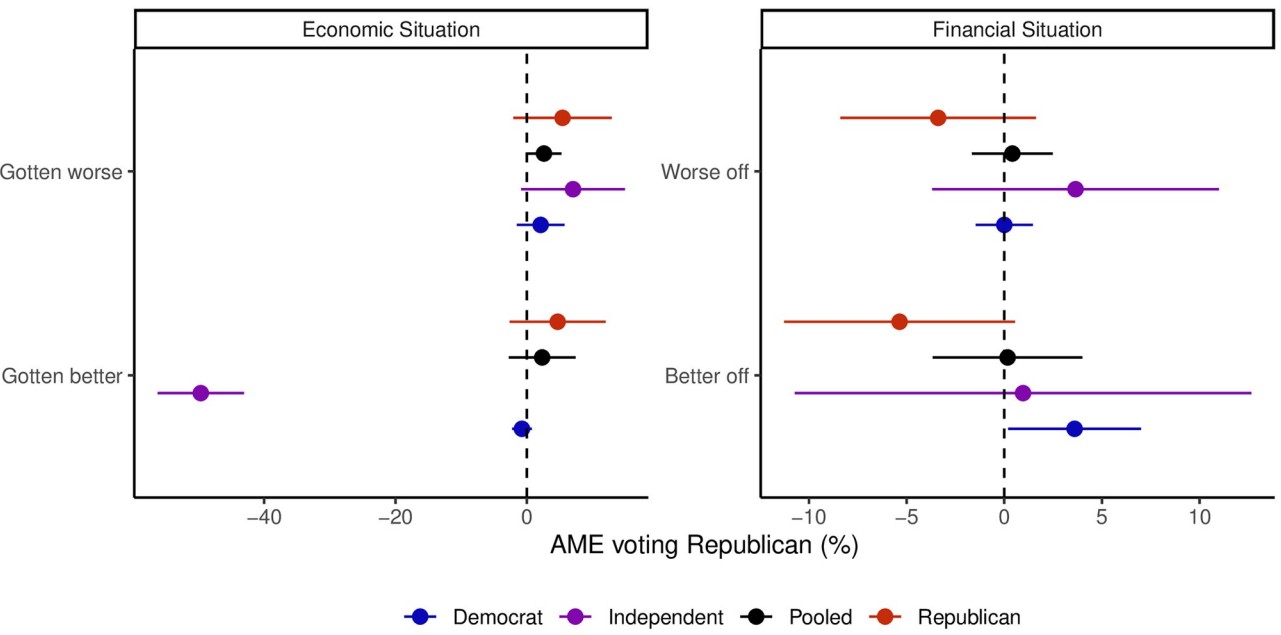

**Fig 5. Economic evaluations' effect on voting Republican for pooled and party-based logit models.**

**Hypothesis**: Should the midterm-as-a-referendum model prove accurate, we expect that voters' evaluations of the President's party will exert a strong influence on their decision to vote against it, particularly among persuadable voters.

To test this hypothesis, we specifically employ two survey questions. The first question addresses voters' evaluations of their personal financial situation, while the second pertains to their assessment of the national economic situation. These questions serve as an indirect means of measuring voters' sentiments towards the incumbent President's party. We favor this indirect approach over direct questioning, as it minimizes the potential for strong correlations with party identification or ideology, leading to a more nuanced understanding of voters' decision-making process.

The findings presented in Fig 5 provide only limited support for the midterm-as-a-referendum model. Specifically, when examining the average marginal effects on voting Republican as function of voters' evaluations of the national economic situation and their personal financial situation, the results tend to converge around zero in both the pooled model and when analyzed by party identification. However, there is an intriguing exception where Independents show a higher likelihood of voting for Democrats when they perceive an improvement in the economy. Nevertheless, as depicted in Fig 3, this view was held by only a small fraction of Independents, indicating a minimal impact on the overall election outcome.

In conclusion, the evaluations of the national economic situation and personal financial circumstances did not significantly predict the preferences of the majority of persuadable voters. This observation remains consistent even among partisans who strongly identify with a specific party. Such evidence directly contradicts the expectations of a midterm-as-a-referendum model, where these factors should exhibit strong predictive power.

### 4.2 Testing the issue-based model

In light of the scant empirical support for the midterm-as-a-referendum model and the complexities involved in testing a balancing model, we redirect our attention to an issues-based model.

**Hypothesis**: If the issues-based model is accurate, we would anticipate that the preferences of persuadable voters will align with their stated belief in the importance of specific issues. Drawing meaningful conclusions and identifying the key issues that explain voters' preferences in these elections will be possible based on these results.

In Fig 6, compelling evidence is presented, indicating that issue importance played a significant role in predicting voters' preferences during this election. The figure shows the average marginal effect of stating the importance of each issue, with data pooled from all voters. While most issues did not show a statistically significant effect, violent crime and foreign policy exhibited a positive relationship with the likelihood of voting for Republican candidates. Conversely, economic inequality and abortion displayed an opposite pattern, with individuals being more inclined to vote for the Democratic candidate if they considered these issues as somewhat or very important. This suggests that, in aggregate, perceptions of issue importance strongly influenced voters' preferences.

The data presented in Fig 4 indicates that a majority of individuals from all political parties considered violent crime, abortion, and foreign policy to be important issues. However, when it comes to economic inequality, Democrats and Independents assigned it a higher level of importance compared to Republicans. These findings suggest that significant shifts in these particular issues may have a broad impact on the entire electorate. This stands in contrast to the test of the midterm-as-a-referendum model, where the significant results only applied to a small subsection of the population.

As discussed in Section 3.3, there are strong indications that voters from different parties perceive various issues with varying degrees of importance. Furthermore, these individuals may respond differently based on their party identification due to several factors, including the prominence of particular issues, internal diversity within parties, and differing opinions on how to address these concerns. Additionally, the way self-identified Democrats and Republicans assess the importance of issues may lead to a net effect of zero, even if these issues were

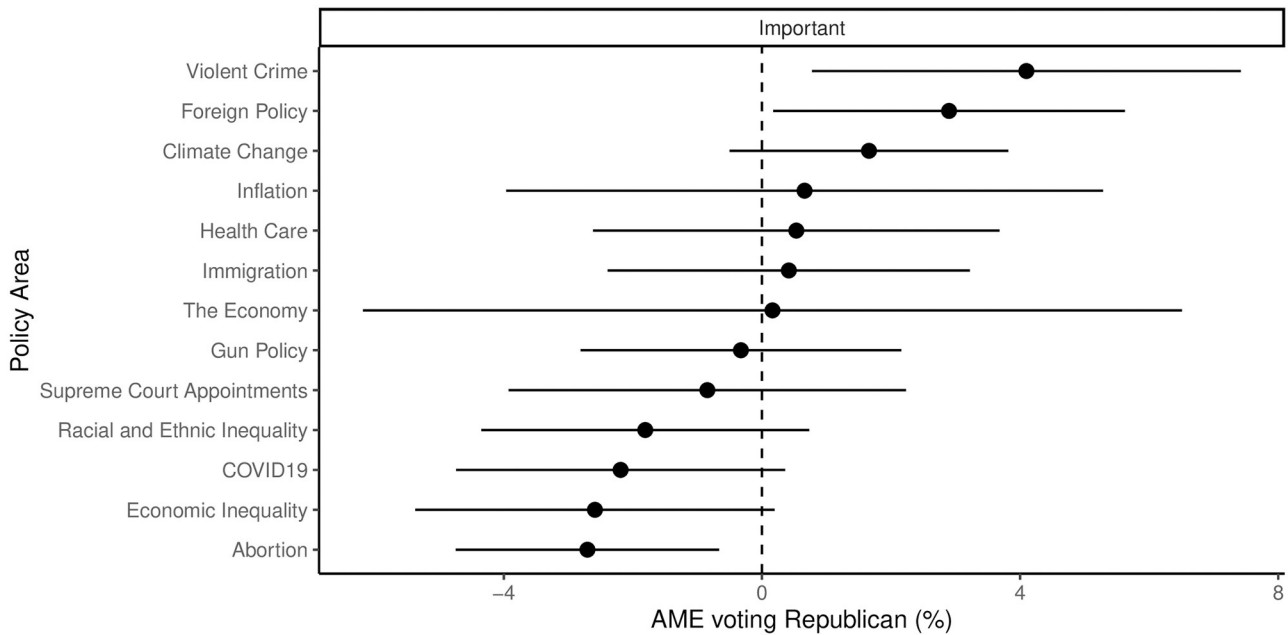

**Fig 6. Average marginal effect (with 95% confidence intervals) of viewing policy areas as important on the probability of voting for the Republican congressional candidate.** The plots show the results for the pooled model.

decisive in influencing their voting behavior. This could be influenced by their preferred media sources, news consumption habits, social networks, and political environments. To address these complexities, we adopt a party-based model to examine the average marginal effects of different issues conditional on their party identification.

Fig 7 provides a visual representation of the average marginal effects for Independents. Since partisanship strongly influences vote choice, our analysis focuses primarily on Independent voters, as they constitute the most persuadable segment of the electorate. Our objective is to ascertain whether specific issues played a predictive role in shaping their party preferences during the election. For a comprehensive view of the results, encompassing all parties and the pooled model, please refer to Fig A in S1 Appendix.

Upon comparing the impact of issue importance on vote choices, it becomes evident that the effects on partisans are smaller compared to Independents. Notably, no significant issue effects are observed for Democrats. However, among Republicans, we identify slight Republican-biased effects concerning foreign policy, and on the other hand, we find small Democrat-biased effects related to abortion and economic inequality.

Among Independent voters, we find compelling evidence in support of an issues-based model of the election. As depicted in Fig 7, we explored several issues in our survey. Notably, violent crime and abortion continue to serve as predictors of how Independents voted in the election, with significantly larger effect sizes compared to the pooled model. In the pooled model, the estimated average marginal effect for violent crime stands at 4.1%, whereas in the Independent party-based model, it increases to 23.9%. Similarly, in the pooled model, abortion exhibits an average marginal effect of −2.7%. However, among Independents, the average marginal effect for abortion becomes −12.3%. This ten percentage point increase in the effect size provides meaningful evidence supporting the notion that abortion significantly influenced voters' electoral choices. Interestingly, the results are not contingent on explicitly stating which party is preferable on the issue. The mere fact that Independents found the issue important was strongly correlated with voting for Democrats.

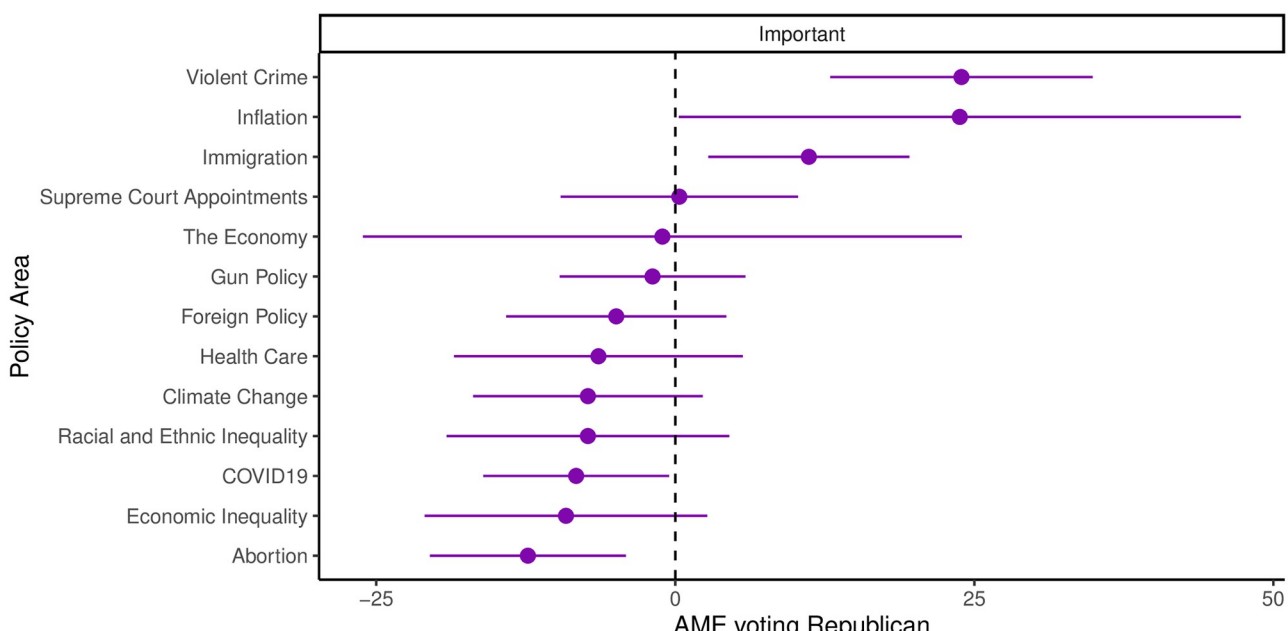

**Fig 7. Average marginal effect (with 95% confidence intervals) of viewing policy areas as important on the probability of voting for the Republican congressional candidate.** The plots show the results for the party-based model for Independents.

Among Independents, if they indicated that violent crime and immigration were important issues, they were considerably more inclined to vote Republican, holding all other factors constant. However, when examining the three economic issues—inflation, the economy, and economic inequality—the average marginal effects reveal either significant variance (inflation), no effects (economic inequality), or a combination of both (the general state of the economy). These findings of null effects and high variance align with our earlier results in Section 4.1, which indicated that economic evaluations played a minimal explanatory role in this election.

## 4.3 2020 presidential benchmark

Considering the notable influence of abortion, violent crime, and immigration on the November 2022 midterm election results, we seek to determine if these findings remain consistent across previous elections. The unexpected shift towards Democrats in this particular election has drawn our attention to the electorate's response to the abortion issue, which emerged as a significant factor influencing Democratic House vote choices. Our investigation will center around testing the validity of the claim that the *Dobbs* decision played a role in amplifying the importance of abortion in the minds of voters. We intend to measure a reduced-form estimate of the magnitude of this change and comprehend how it affected voting patterns. Concurrently, we will explore whether the responses to the issues of violent crime and immigration remained stable during the same period.

If we observe an unusually strong impact of abortion on voters' preferences, indicating that the recent ruling has elevated the importance of the abortion issue among Independent voters, it could have substantial implications for issue-based models. Such findings may provide valuable insights into how issues gain significance and how persuadable voters shape their party preferences in response to critical policy changes. To delve deeper into this matter, we capitalize on the exogenous variation presented by the end of *Roe v. Wade* and examine how the importance of the abortion issue changes in reaction to a major policy shift that falls beyond the direct control of Congress.

The 2022 midterm elections present a unique opportunity to explore how external shocks impact voter behavior and disrupt traditional patterns observed in such elections. In June, prior to the November 2022 election, the Supreme Court's decision to overturn 50 years of Federal legal protection for abortion marked a significant policy change over which neither the Democratic Congress nor the Democratic President had direct control. With the exception of the Senate's role in selecting Supreme Court judges over the previous 50 years, the legislature had limited influence over the specifics of this decision. Before 2022, many Democrats and Republicans assumed that *Roe* was settled law.

Given that the *Dobbs* decision was an unexpected policy change outside the control of both the President and Congress, it provides a clear opportunity to observe how this change affected voter behavior, particularly through the lens of increased issue importance. To measure the impact of this change on the predictive effect of abortion as an issue, we establish a historical benchmark by comparing the 2022 electorate with that of 2020, utilizing the same model as in Section 4.2. This comparison enables us to gauge the magnitude of the change in voters' responses to the abortion issue following the *Dobbs* decision.

By comparing the opinions of the 2022 electorate on various issues with the opinions of the 2020 electorate, we aim to isolate and understand how shifts in opinions on abortion, violent crime, and immigration correlate with vote choices between the two elections. If the magnitudes of these shifts are similar for persuadable voters in both elections, it would indicate that abortion, violent crime, and immigration consistently remained important issues. In such a scenario, while we may not be able to entirely rule out the possibility that the *Dobbs* decision

influenced how voters perceive the issue of abortion, we would not have evidence to support that it caused a change in voters' perceptions. On the other hand, if the magnitudes of the average marginal effects for immigration and violent crime remain consistent between the two elections, while abortion shifts from having no predictive power to becoming a significant predictor of vote choice, it would provide evidence consistent with the notion that the Supreme Court's actions in *Dobbs* indeed changed the perceived importance of abortion among voters. The graph depicted in Fig 8 provides a visual representation of the average marginal effect on voting for the Republican candidate as opposed to the Democratic candidate when shifting from considering an issue as insignificant (either not too important or not important at all) to perceiving it as significant (somewhat important or very important). The three main issues we have identified as potentially relevant are examined in this context. By comparing the

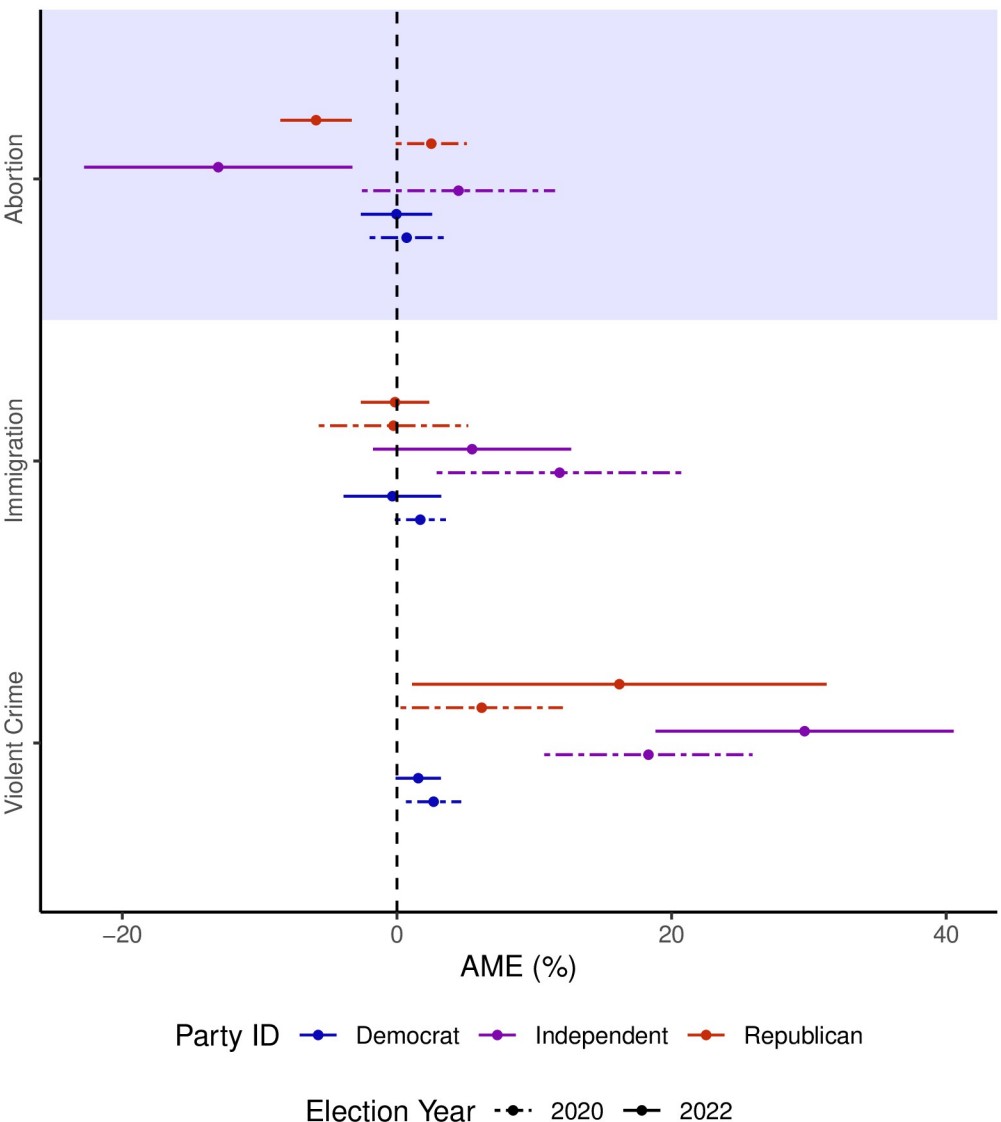

**Fig 8. Average marginal effect from thinking abortion, immigration, and violent crime were important on the probability of voting Republican for the 2020 and 2022 surveys.** There is a clear break in the relationship between importance attributed to abortion and voting decisions for Independents and Republicans. There is no such break for any partisans for immigration or violent crime.

coefficients in the models, we can ascertain whether there have been any changes in the estimated importance of these issues in voters' decision-making processes.

The influence of the *Dobbs* ruling becomes more evident when examining the average marginal effects on voter preferences regarding the importance of the abortion issue. In the 2020 election, the average marginal effect of abortion on voting Republican showed no significant impact for both Independents and Republicans. The marginal effect for Independents was 4.5%, while for Republicans, it was 2.5%. Although these effects slightly favored the Republicans, the magnitudes were small and not statistically significant, suggesting that abortion had minimal influence on vote choices during that time. In stark contrast to the 2020 benchmark, the marginal effects in the 2022 election for the abortion issue exhibit substantial changes. The effects for both Independents and Republicans grew significantly to −13% and −6%, respectively. This shift indicates that abortion now favored Democrats for both groups, and the results are statistically significant. These large values suggest that the abortion issue played a significant role in the 2022 election.

On the other hand, when analyzing the marginal effects for violent crime and immigration between the two years, they largely overlap. In both 2020 and 2022, Independents were slightly more inclined to vote for Republican candidates if they perceived either violent crime or immigration as somewhat or very important issues. The magnitude of these effects remains consistent between the two election years. This consistency suggests that violent crime and immigration were not responsible for the unexpected nature of the 2022 midterm election.

Undoubtedly, the findings unequivocally demonstrate a noteworthy increase in the significance of the abortion issue among both Republicans and Independents during the period spanning from 2020 to 2022. This change in perception regarding abortion seems to be one of the key factors contributing to the substantial shift in favor of the Democratic Party during the 2022 midterm elections. By comparing these findings with a similar model from the 2020 election, not only do we authenticate the importance of the observed changes, but we also eliminate the influence of other matters like immigration and violent crime that might have otherwise obscured the true impact of the *Dobbs* ruling on the election outcomes.

## 5. Discussion and conclusion

Prior to the 2022 midterm elections, traditional models used to forecast such events pointed towards significant losses for the Democratic Party. Several factors contributed to this outlook, including a slowdown in economic growth, a surge in inflation, and declining approval ratings for President Joe Biden, all of which indicated potential challenges for the President's party in retaining its slim majorities in both houses of Congress. Well-established academic frameworks like the midterm-as-a-referendum and midterm balancing models further reinforced the notion that there was a considerable risk of the President's party losing ground in the upcoming elections. As a result, experts and scholars were generally convinced that the Democratic Party was poised to suffer defeats similar to what the President's party experienced in the 2018, 2014, 2010, and 2006 midterm elections.

However, the actual election results caught pundits and scholars off guard. The Democratic Party only lost nine seats in the House of Representatives and managed to gain one seat in the Senate, a performance on par with the incumbent party's showing in the 2002 elections following the 9/11 terrorist attacks and the 1998 elections following Bill Clinton's impeachment. Moreover, Democrats achieved a breakthrough by taking control of Michigan's legislature, marking the first time the President's party had increased the number of state legislatures under its control since 1934.

This paper conducts a thorough analysis and evaluation of various theories concerning voter behavior in midterm elections. Our analysis indicates that the midterm-as-a-referendum model falls short in explaining the 2022 data. Surprisingly, voters' electoral choices showed little correlation with their perceptions of the national economic situation and their personal financial circumstances. Although the balancing model, with its complex strategic assumptions about voter behavior, proves challenging to disprove, its main predictions do not align with the actual outcome of the midterm elections.

In light of these challenges, our approach focused on investigating the predictions of an issue-based model, where voters base their voting decisions on the issues they deem significant. The results of our study demonstrate a robust correlation between voters' beliefs about the importance of specific issues and their voting behavior. Notably, issues such as abortion, crime, and immigration exerted a significant influence on the voting choices of Independent voters during the midterm elections, particularly when they perceived these issues to be crucial. In summary, our analysis strongly supports the idea that an issue-based model aligns best with the available data.

Our analysis presents compelling evidence of the significant impact the *Dobbs v. Jackson Women's Health Organization* case had on raising the prominence of the abortion issue. On May 2nd, 2022, just six months before the midterm elections, *Politico*, a prominent political news website, leaked a draft of the Supreme Court's decision for this case. The leaked draft, which ultimately resulted in the overturning of the landmark *Roe v. Wade* ruling, captured widespread public interest. When the official decision was released on June 24th, the public's attention was further drawn to the matter. In the wake of the decision, swift reactions from politicians followed. On the one hand, states such as California, Colorado, and Vermont introduced state-level constitutional amendments to safeguard the right to abortion. On the other hand, thirteen states had trigger laws in place, which would automatically enforce abortion bans if *Roe v. Wade* were overturned. As a result of this pivotal development, a wave of rallies, state-level referenda, and extensive news coverage ensued, all dedicated to passionate debates surrounding the contentious issue of abortion.

In the lead-up to the midterm elections, Republican leaders asserted that abortion would not be the determining factor for voters, emphasizing that crime and inflation would hold greater significance in the minds of the electorate. Nevertheless, our research uncovers a different reality, demonstrating that abortion, alongside crime and immigration, played a vital role in shaping the decisions of Independent voters. Particularly noteworthy was the substantial increase in the impact of the abortion issue compared to previous elections. In contrast, inflation emerged as a less dependable predictor of Independent voters' choices, offering only ambiguous signals in comparison to the other issues.

Following the rescission of *Roe v. Wade*, voters' perception of abortion as an important issue comes as no surprise. However, relying solely on cross-sectional data does not definitively prove that abortion played a more decisive role in the 2022 midterm elections compared to previous ones. To confidently attribute the *Dobbs* decision's influence on the 2022 midterms, it is crucial to establish whether abortion had consistently been a decisive factor in the past. It is possible that abortion, alongside crime and immigration, has consistently held pivotal importance in voters' decision-making. Yet, our surveys present compelling evidence to the contrary. We found that the impact of the *Dobbs* decision led to a notable increase in the correlation between abortion's perceived importance and voting behavior. While other policy matters, such as crime and immigration, remained significant among Independent voters, their influence on voting behavior showed little change compared to the 2020 elections. In contrast, the significance of stating abortion as a somewhat or very important issue witnessed a substantial shift. This factor went from a 4.5% advantage in favor of Republicans

in 2020 to a 13% advantage in favor of Democrats in 2022, indicating a significant structural break.

This change suggests that issue importance can be significantly influenced by sudden, dramatic policy changes with far-reaching consequences. Such changes have the potential to overshadow other concerns for voters and bring a new issue to the forefront, resulting in a structural break in the political equilibrium. While the Republican Party's base strongly opposes abortion, the broader electorate holds more moderate views on the matter. When Republicans campaigned on restricting the rights to abortion for individuals with uteruses, many persuadable and cross-pressured voters remained unconvinced. The presence of *Roe v. Wade* provided a sense of security for these rights, making the issue somewhat tangential to Independent voters' electoral behavior. However, as the policy environment shifted due to the Supreme Court's *Dobbs* ruling, voters had to re-evaluate their preferences. It appears that pivotal voters cast their ballots with abortion in mind, recognizing its newfound significance after the *Dobbs* decision. This ruling led to a recalibration of priorities and prompted voters to consider abortion as a more central issue in their electoral decision-making.

In conclusion, this paper demonstrates the potency of an issues-based model for comprehending and elucidating midterm elections. When viewed through the issue-based lens, the results of the 2022 midterm elections were neither surprising nor inexplicable. Unlike alternative theories, the issue-based model presents verifiable predictions even in the face of significant policy shocks. We firmly believe that this framework offers a valuable roadmap for future research on electoral behavior. Just as it proved beneficial in analyzing the 2022 midterm elections, this framework can be utilized to formulate testable hypotheses regarding key issues in an election, identify structural breaks in political dynamics, and provide falsifiable explanations that align with unexpected outcomes and electoral context.

## Supporting information

**S1 Appendix. Supplemental information, question wording, Figures, and Tables.**
(PDF)

## Author Contributions

**Conceptualization:** Claudia Kann, Daniel Ebanks, Jacob Morrier, R. Michael Alvarez.

**Data curation:** Claudia Kann, Daniel Ebanks, Jacob Morrier.

**Funding acquisition:** R. Michael Alvarez.

**Investigation:** Claudia Kann, Daniel Ebanks, Jacob Morrier, R. Michael Alvarez.

**Methodology:** Claudia Kann, Daniel Ebanks, Jacob Morrier, R. Michael Alvarez.

**Project administration:** R. Michael Alvarez.

**Software:** Claudia Kann, Daniel Ebanks, Jacob Morrier.

**Validation:** Claudia Kann, Daniel Ebanks, Jacob Morrier.

**Visualization:** Claudia Kann, Daniel Ebanks, Jacob Morrier.

**Writing – original draft:** Claudia Kann, Daniel Ebanks, Jacob Morrier, R. Michael Alvarez.

**Writing – review & editing:** Claudia Kann, Daniel Ebanks, Jacob Morrier, R. Michael Alvarez.

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
