## [Decision Letter · Decision Letter 0]

14 Jun 2023

PONE-D-23-07439Persuadable voters decided the 2022 midterm: Abortion rights and issues-based frameworks for electionsPLOS ONE

Dear Dr. Kann,

Thank you for submitting your manuscript to PLOS ONE. After careful consideration, we feel that it has merit but does not fully meet PLOS ONE’s publication criteria as it currently stands. Therefore, we invite you to submit a revised version of the manuscript that addresses the points raised during the review process.

I have carefully reviewed the paper along with the feedback from the reviewers.

Based on the reviewers' comments and we have reached a decision regarding the publication of your manuscript.

In short, the Reviewer 1 recommends rejecting the paper, citing several concerns. He/She question the connection between public opinion data and congressional seat changes, emphasizing the role of redistricting in understanding such patterns. Additionally, they raise concerns about the inclusion of leaners with true independents, potentially leading to an inaccurate representation of the voter groups. The reviewer also suggests considering other important issues and the threats to democracy in the analysis. Furthermore, they provide minor quibbles and suggestions for improvement, such as adding percents on the Y-axis of Figure 2.

The Reviewer 2, on the other hand, recommends accepting the paper. He/She acknowledge the detailed analysis of the 2022 midterm elections and the well-supported conclusion regarding the role of abortion as a salient issue. While they suggest discussing the limitations of the study, such as sampling bias and non-response bias, they appreciate the comprehensive analysis of the factors influencing the election. Reviewer 3 provides a major review and raises important concerns. He/she caution against making strong claims about issues explaining the election outcome and suggest conducting a counterfactual simulation to assess the impact of abortion and violent crime on the percentage of voters voting Democrat. He/She also suggest revisions to clarify the hypotheses tested, streamline the text, and improve the transparency of the empirical strategy. Reviewer 4 also provides a major review, highlighting the novelty of the study and suggesting improvements. He/She recommend including a literature review on strategic voting behavior, clarifying the hypotheses tested, and presenting and discussing the results after testing the hypotheses. The reviewer also suggests making the empirical strategy more transparent, revising the density of the text, and addressing minor formatting issues.

While the reviewers appreciate the relevance and potential contributions of your study, they have raised valid concerns and provided valuable suggestions for improvement. I agree that addressing these concerns and implementing the recommended revisions will strengthen your manuscript and make it suitable for publication in our journal. Please carefully consider the reviewers' comments and revise your manuscript accordingly. My suggestion is to address, specially, the following issues in your revised submission:

1) Clarify the connection between public opinion data and congressional seat changes, considering the role of redistricting as a major factor.

2) Revisit the inclusion of leaners with independents, ensuring the appropriate identification of voter groups.

3) Consider including other important issues and threats to democracy in the analysis, as suggested by Reviewer 1.

4) Discuss the limitations of the study, such as sampling bias, non-response bias, self-reported bias, timing bias, and limitations of weighting, as highlighted by Reviewer 2.

5) Conduct the suggested counterfactual simulation to assess the impact of abortion and violent crime on the percentage of voters voting Democrat, as recommended by Reviewer 3.

6) Provide a clearer literature review on strategic voting behavior, as suggested by Reviewer 4.

7) Streamline the text to improve clarity and highlight the hypotheses tested, addressing the concerns raised by Reviewer 4.

8) Make the empirical strategy more transparent by presenting the model equation and commenting on the estimation results, as recommended by Reviewer 4.

9) Address minor formatting issues pointed out by Reviewer 4.

Ensure that your revised submission includes a detailed response to each reviewer's comments, indicating the specific changes you have made in the manuscript. Additionally, provide a response letter that summarizes the revisions made and highlights how you have addressed the concerns and suggestions raised by the reviewers. This will help the editor and reviewers evaluate the extent to which you have addressed their feedback and revised the manuscript accordingly.

We look forward to receiving your revised manuscript.

Kind regards,

Carlos Henrique Gomes Ferreira, Ph.D.

Academic Editor

PLOS ONE

Journal Requirements:

Reviewers' comments:

Reviewer's Responses to Questions

**Comments to the Author**

1. Is the manuscript technically sound, and do the data support the conclusions?

Reviewer #1: No

Reviewer #2: Yes

Reviewer #3: No

Reviewer #4: Yes

2. Has the statistical analysis been performed appropriately and rigorously? 

Reviewer #1: Yes

Reviewer #2: Yes

Reviewer #3: Yes

Reviewer #4: Yes

3. Have the authors made all data underlying the findings in their manuscript fully available?

Reviewer #1: Yes

Reviewer #2: Yes

Reviewer #3: Yes

Reviewer #4: No

4. Is the manuscript presented in an intelligible fashion and written in standard English?

Reviewer #1: Yes

Reviewer #2: Yes

Reviewer #3: Yes

Reviewer #4: Yes

5. Review Comments to the Author

Reviewer #1: This is an interesting paper using cross sectional data to examine correlates of voting behavior in 2022 & 2020. The authors ask an important question about the 2022 election and whether issues mattered to voters and whether that impacted the number of seats won in Congress by each party.

There are number of problems with the current manuscript that need to be considered before it is publishable. Given the findings and the methods design, this article would be more suitable for APR or Political Behavior.

One general problem is that the paper wants to say something about the rather small seat changes in the US House, but is inferring that because abortion appears to have mattered to some voters across the country that it resulted in the pattern we see at the aggregate level. There is no way we can conclude from the results shown here that this caused the seat change pattern we observed or are even correlated with it. In particular, a seat change model would need to consider the massive gerrymandering going on across the country that must be a large factor in understanding Congressional seat change this cycle. Almost certainly Florida, New York, Illinois, etc changes in redistricting were major contributors to what happened in Congress. Here’s an interesting take: https://fivethirtyeight.com/features/redistricting-house-2022/). But regardless, there’s no way we can connect the public opinion data in this paper to what happened to congress without a serious analysis of redistricting, which is a major exogenous shock to the system.

Another problem is that the authors use the 3 point party id scale, throwing leaners in with true independents. Yet, we know from an abundance of literature that leaners appear to be actually stronger partisans then weak partisans. More weak partisans are likely to defect than leaners. Throwing leaners into the mix of independents is really a problem because in truth many “independents” are really partisans. Thus, the authors have not isolated the appropriate group of voters. They’ve created a large box of voters, most of whom are actually partisans.

In addition, why would abortion shape the federal races given that this was thrown back to the states. I think the authors are probably right abortion did matter in key races –maybe Arizona (for GOV), maybe Michigan (for GOV) (almost for sure with it also on the ballot), maybe Pennsylvania (for SEN) but these were at the state level. I think making this important at the congressional level where incumbency reelection is so high already and information is so low is a hard sell in the face of other issues –election deniers for example, which has been another popular spin on issue importance. I notice the authors did not tap this dimension or the broader threats to democracy on their survey, which seemed to have been a popular issue theme for both Democrats and Republicans.

I have a number of minor quibbles that the authors should also consider. While it is true that some models predicted a Senate switch, this was not universal. Other conventional wisdom suggests that the president is likely to hold onto the Senate in their first term in the midterm.

Maybe add percents or something on the Y axis to Figure 2.

The authors say that the winners weren’t “declared” for a week after the election. Maybe in some states, but even so most congressional races were called almost immediately. Re-election rates in Congress were 98% and typical and most of those were not competitive. While a few close races were called much later I think arguing that such timing reduces bias for the winner in the data is not likely.

There is a large literature on the most important issue question. The authors need to engage that literature especially as it relates to vote choice.

I found it quite surprising that foreign policy was important in the 2022 election and led to greater support for Republicans. I’m just not sure what to make of that.

Reviewer #2: The paper provides a detailed analysis of the 2022 midterm elections, highlighting the discrepancy between the traditional models of midterm elections and the actual election results. The paper suggests that the issue-based model is most consistent with the data gathered from a nationally representative survey of U.S. registered voters. The study further reveals that the Supreme Court’s decision in the Dobbs v. Jackson Women’s Health Organization case played a crucial role in raising the salience of abortion. The conclusion is well-supported by the survey data and provides a plausible explanation for the election results. The paper could have benefited from a more detailed discussion of the limitations of the study, such as the scope of the survey and the potential biases in the sampling process. However, overall, the conclusion is well-argued and provides a comprehensive analysis of the factors influencing the 2022 midterm elections.

Here are some comments on the methodology:

Sampling bias: The survey was conducted online and relied on opt-in survey subject panels. This method of sampling may not represent the entire population, as not everyone has internet access or chooses to participate in surveys.

Non-response bias: Even though the authors attempted to reduce the effects of non-response bias by offering compensation and weighting by various demographic factors, it is still possible that those who choose to respond to surveys are not representative of the population.

Self-reported bias: The authors used self-reported data to measure economic and national economy evaluations, which may be subject to bias.

Timing bias: Although the authors attempted to reduce the effects of media narratives by collecting data in the days immediately following the election, it is still possible that media coverage and other external factors could have influenced respondents' answers.

Limitations of weighting: The authors used weighting to adjust for various demographic factors, but this method may not fully address the potential biases in the sample.

Limited generalizability: The survey was conducted in November 2022, immediately following the 2022 midterm election, and may not be generalizable to other time periods or elections

Reviewer #3: Summary: In this paper, the authors propose an issue-based model for voting preferences among voters. They apply this assumption to a poll from 2022 and 2020 to see how abortion became a more important issue in 2022, and this was more likely to drive users to vote for Democrats. The results demonstrate that the a referendum model of voting is insufficient to explain voter behavior, while the marginal effects of issues had a substantial impact on voting, especially among independents.

Major comments:

- While I am convinced that issues are important correlates to voter preference, I would be cautious to say that issues explain the election outcome. When the authors said strong statements, such as “Persuadable voters decided the 2022 midterm” I took pause becayse the data does not necessarily agree with this statement. Instead, even if independent's choices changed, we do not know the number of independents who were inspired to vote because of the abortion debate. In other words, the Dobbs decision's impact could have been a stronger motivation for Democrats to vote, and the change in the votes from the Independents who actually voted could have been minimal (I would be more convinced had voting been a requirement, as it is in many countries, in which case motivation would not have been as big a factor). I do not think this is a fundamental flaw, but rather something that is worth mentioning, and therefore I would be cautious to use as strong a title as the current one.

Another critique is the idea that abortion was the deciding issue, given a similar effect size was seen for violent crime. How can we say that abortion drove a change but violent crimes did not cancel out this effect, given it too was a larger factor in 2022 than 2020? If both factors were equally important, then they would cancel each other out, and the reason for the Democrats' win would be presumably higher motivation for Democrats to vote than Republicans. To address this fundamental problem, I suggest the authors create a simulation to show a counter-factual. Cf.

Daniel Hickey, Matheus Schmitz, Daniel Fessler, Paul Smaldino, Goran Muric, Keith Burghardt. (2023). No Love Among Haters: Negative Interactions Reduce Hate Community Engagement. arXiv preprint arXiv: 2303.13641

The counter-factual will utilize the model the authors already created; the actual effort to create this counter-factual is relatively minor.

1. In the present paper, the authors could show the simulated % of voters voting Democrat in 2022. This is the logic model applied to each datapoint. One datapoint with features X1 will have a probability p1 of voting Democrat. Another datapoint with feature X2 will have probability p2 of voting democrat. The expected number to vote Democrat will be p1+p2. After applying the model to each datapoint and summing all the probabilities together, you will have the expected percent of voters who will vote Democrat: (p1+p2+p3+...)/(number of people polled). This percentage can be compared with the actual data - e.g., you will predict 55% of voters will vote Democrat, but the data shows a 54% percent. This will indicate that the model correctly estimates who votes for whom.

2. Next, they can compare this model to one in which all parameters are the same but the coefficient for abortion = 0 (i.e., abortion were forced to have no effect on voting preferences), and similarly a simulation where all coefficients are the same except violent crime is set to 0. This will adjust each datapoint's probability to vote Democrat. Taking the sum of probabilities will create the appropriate counter-factual condition. We should see:

• <50% of voters voting Democrat if abortion coefficient is set to zero

• >50% of voters voting Democrat if the violent crime coefficient is set to zero (and the absolute change compared to status quo is minimal)

The reason I argue this is an important contribution is again because violent crime and abortion are both major issues. If the impact of violent crime was small in an absolute sense then removing that coefficient would not strongly impact the percent of voters voting Democrat. Alternatively, if the abortion importance coefficient were set to 0, we might expect the % of voters voting Democrat to change dramatically. In contrast to the covariates extracted from your model, this simulation accounts for the relative number of people who are Democrats, Independents, and Republicans, as well as the number who consider each issue important.

In summary, I believe the paper is quite good. My critiques are:

1. That the claims they make are strong given the alternative scenarios that cannot be accounted for with the present data, and

2. The current paper lacks the exemplar convincing figure or table that shows what the authors argue: had abortion not been a deciding factor, the Democrats would have lost. If the above simulation is feasible, and I believe it is, this could be a single figure that, in my opinion, will decisively show just that.

Reviewer #4: I read the paper with great interest. The authors argue that the 2022 midterm elections had unpredictable results. Using a sample of nationally representative registered voters surveyed shortly after the November 2022 midterm elections, the authors found that factors other than the fundamentals were responsible for the election outcome, as abortion appeared to be an important and highly salient issue in that election.

The political system appears to have suffered an exogenous shock from the Supreme Court's decision to overturn Roe v. Wade in June 2022, so abortion was seen as a key issue in this election campaign

The issue is certainly of interest and opens up many possibilities for action in the next rounds of elections.

My general impression of the paper is that this study opens an interesting perspective on other factors that can influence the outcome of election campaigns. On the other hand, there are parts of the paper that, if revised, would bring more clarity to the arguments.

I would recommend;

• A section on the literature review on strategic voting behavior.

• The text is very dense and can be shortened somewhat. In my opinion, the paper could benefit from highlighting the hypotheses tested (explicitly stated) so that the reader can easily understand the procedure chosen by the authors.

In my opinion, the paper could benefit from presenting and commenting on the results after testing the hypotheses

• Empirical strategy : The model specification is very intransparent. The reader wants to take a look at the estimates and at the figures and tables and needs to understand immediately what it is all about. I suggest that the authors first write down the equation of the model to be estimated and then present the estimation table, which needs to be commented on right after. This will make the empirical strategy more transparent. Then, I would suggest describing the most basic model and later come up with a more complicated equation or just verbally state what other terms are added to this basic model. In general, I do not understand exactly what these estimates mean because of the intransparency presentation of the estimates. Therefore, it is difficult for me to comment on the interpretation of the results later. However, I like the graphical representation of the results.

MINOR:

1. In the third last line on p.4 : "Organization decision, which held that the the Constitution does not provide a right to" there is an additional "the"

2.

3. The regression table D needs a little reformatting because the title and the columns do not match

6. PLOS authors have the option to publish the peer review history of their article (what does this mean?). If published, this will include your full peer review and any attached files.

Reviewer #1: No

Reviewer #2: No

Reviewer #3: No

Reviewer #4: No

---

## [Author Response · Author response to Decision Letter 0]

16 Aug 2023

PLEASE NOTE, WE HAVE SUBMITTED A FORMATTED VERSION OF THESE RESPONSES IN "RESPONSE TO REVIEWERS.pdf". 

INCLUDED BELOW FOR COMPLETENESS

Dear Editor,

Thank you for your detailed comments and for providing us with four useful external reviews. We start this memo by summarizing what we have done in response to your nine revision requests. We then discuss the other formatting changes we have made to our paper, per your requests. Thereafter we provide a point-by-point response to the questions, comments, and concerns raised by the four reviewers.

We appreciate the opportunity to revise and resubmit our research to PLOS ONE. We believe that this revision process has greatly clarified our research and has strengthened our work. 

Below we have left your comments and those of the reviewer in black text. Our responses are in red text. 

Sincerely,

The Authors

Responses to the Editor:

1) Clarify the connection between public opinion data and congressional seat changes, considering the role of redistricting as a major factor.

We thank R1 for their comments about redistricting. We have tried to clarify in the text that our paper is not about modeling seat swings, nor about explaining aggregate vote shares across congressional districts. Rather we are simply using the pre-election punditry to help motivate our analysis of our individual-level survey data from the 2020 and 2022 elections. We also added a paragraph on this point in our Discussion. 

The paper is about individual level voting behavior in the 2022 midterm elections, 

We appreciate R1’s reference to the 538 blog about the post-2020 redistricting, and we note that there are plenty of other reputable blogs that argue that Republicans would benefit from their gerrymandering efforts (examples are the Brennan Center [https://www.brennancenter.org/our-work/analysis-opinion/after-redistricting-heres-how-each-party-could-win-house], the Guardian [https://www.theguardian.com/us-news/ng-interactive/2021/nov/12/gerrymander-redistricting-map-republicans-democrats-visual] and the New York Times [https://www.nytimes.com/2022/09/30/upshot/midterms-gerrymandering-republicans.html]. Our reading of the pundits is that they mostly argue that the Republicans gave themselves an edge going into the 2022 midterms because of redistricting, but as the New York Times points out it was not insurmountable. 

We also would like to note that any partisan advantage that the pundits believed that redistricting would provide before the 2022 midterms would be factored into the pundit estimates for which party would do best in the 2022 midterm election. In this sense, gerrymandering is taken into account in our discussion about how the pundits viewed the 2022 midterm election. 

But to reiterate, our analysis is trying to explain individual-level voting decisions in the 2022 midterm elections, what factors drove voters to vote Republican or Democratic, and how the relative contributions of those factors to voting decisions may have changed between 2020 and 2022. We are not studying vote shares nor seat shifts between elections, nor how votes translate into seats. Those are important questions about contemporary American politics, but they are outside of the scope of the research reported in our paper. 

2) Revisit the inclusion of leaners with independents, ensuring the appropriate identification of voter groups.

We thank Reviewer 1 for noting this important point regarding how we classify voters’ party identification. To address this point, we have re-run our analysis following the convention suggested by the reviewer to show it does not change the topline results of the paper. In this re-analysis,, Republican and Democratic leaners are grouped with their respective parties. For this exercise, only voters who indicate they are independents are classified as such. We show that these changes have only a marginal effect on the topline results.

First, in SI Fig F.2, the share of Democrats and Republicans voting for their respective parties does not meaningfully change from Fig 2. There is, however, a large shift in how the Independents vote — The margin moves from 46-53 in favor of Republicans to 40-59. That is, when excluding leaners, Independents appear significantly more Republican. This means that under this recoding, Independents who are the most persuadable voters strongly preferred the Republican candidate for Congress. 

Second, in SI Fig F.3, we decompose congressional vote preferences by economic and personal financial perceptions. In both cases, the results map to Fig 3 in the main text, with minimal differences. 

Third, in SI Fig F.4, we show that Independent voters are slightly more likely to vote for a Republican in Congress across issues, but this is not surprising given that the voters classified as independent are more Republican leaning than our initial coding. That said, despite small changes in the magnitude of support for Democrats by issue, the direction of how independents voted conditional on the perception of an issue being important is unchanged. 

Next, we replicate the results for Independents probability for voting Republican in SI Fig F.5. Along this dimension, three economic issues are statistically significant – Economic Inequality, the Economy, and Inflation. Stating that abortion is an important issue remains statistically significant and at a slightly larger magnitude (17.5%) compared to the original figure (12.5%). 

Finally, we replicate our main counterfactual in SI Fig F.6. The results for Abortion and Violent Crime are similar to the original text. There is more of a gap between 2020 and 2022 Independents on Immigration, but the Confidence Intervals overlap and the difference is not statistically significant at traditional levels. 

Taken together, although the sample of Independents excluding leaners is 10 percentage points more Republican, the main thrust of the results holds, including the counterfactual in SI Fig F.6. 

3) Consider including other important issues and threats to democracy in the analysis, as suggested by Reviewer 1.

We appreciate Reviewer 1’s comment. While we would like to include many other issues in our analysis, we are limited by what was included in our survey. Our survey did not include issues like election-denialism and threats to democracy. We discuss this in the discussion section, along with the other limitations suggested by Reviewer 4 (see below). We point out in particular that while it would be interesting to include other issues like these in our analysis, as they are unlikely to be strongly correlated with abortion (which is the central issue variable in our analysis) their omission is unlikely to produce bias in our results. We also note that recent unpublished work (which we cite in the Discussion section) found that election-denialism had only a very slight effect in the 2022 midterm; Malzahn and Hall (2023) summarize their results in the abstract of their working paper by saying that the effect of election-denialism in the 2022 election “is small enough to suggest that only a relatively small group of voters changed their vote in response to having an election-denying candidate on the ballot.” 

4) Discuss the limitations of the study, such as sampling bias, non-response bias, self-reported bias, timing bias, and limitations of weighting, as highlighted by Reviewer 2.

We have addressed the limitations discussed by the reviewers in the Data and Methodology section.

5) Conduct the suggested counterfactual simulation to assess the impact of abortion and violent crime on the percentage of voters voting Democrat, as recommended by Reviewer 3.

We appreciate this suggestion by Reviewer 3. We decided not to include the counterfactual suggested in the new submission. The original model correctly predicts that 50.2% of voters will vote for the Democratic candidate. When the coefficient on abortion importance is set to 0, which is the same as saying no one believed abortion to be important, the new result says 46.4% of the electorate will vote for the Democratic candidate. This is a 3.8% swing in favor of the Republicans. If the coefficient on violent crime is set to 0, 56.4% of the electorate would vote for the Democratic candidate—a 6.2% swing in favor of the Democrats. 

We believe these results, however, are misleading. In looking at the results presented in Figure 8, the average marginal effect of violent crime is consistent between elections while the average marginal effect of abortion changes significantly. The average marginal effect represents the change in probability represented by a change of opinion, and therefore is equivalent to the average change from the coefficient going from its existing value to 0. Thus, even though the counterfactual simulation suggested by Reviewer 3 suggests that violent crime has a larger effect, we are interested in why this midterm was unique. We believe that all of the information desired from this counterfactual can be seen in Figure 8, with the clarity of comparing the results to previous elections. 

Doing this exercise was very helpful in confirming our analysis however, and we have worked to ensure that these connections are made clearer in the text. Thank you for the comment. 

6) Provide a clearer literature review on strategic voting behavior, as suggested by Reviewer 4.

Respectfully, most of the literature on strategic voting does not directly apply to our current work. The overwhelming majority of congressional elections are two-candidate elections, so the concept of strategic voting, which involves voting for a candidate other than one's favorite to improve the chances of defeating a less-preferred candidate, is not applicable. While we do discuss sophisticated voting models like policy balancing in our paper, our focus is on Congressional elections, which typically involve only two major candidates — one Democratic and one Republican. As such, the opportunity for strategic voting by voters is minimal. A strategic vote in a two-candidate election would essentially mean voting for someone who is the voter's least favorite candidate, which is never an optimal strategy. While the literature on strategic voting may have relevance in the context of primary elections, where more than two candidates compete, it is not within the scope of our current analysis. Therefore, we briefly acknowledge this aspect along with other limitations and explain why we consider it irrelevant for our present study.

7) Streamline the text to improve clarity and highlight the hypotheses tested, addressing the concerns raised by Reviewer 4.

8) Make the empirical strategy more transparent by presenting the model equation and commenting on the estimation results, as recommended by Reviewer 4.

9) Address minor formatting issues pointed out by Reviewer 4.

We have addressed the minor formatting issues that Reviewer 4 noted. 

We thank the editor for your excellent summary of the important revisions that we needed to make to improve our paper.

Finally we have provided a marked-up and unmarked copy of our revisions, dealt with the funding questions, and have made our data and code available in a repository and provided the DOI for the repository. 

Sincerely, 

The Authors

Responses to Reviewers

One general problem is that the paper wants to say something about the rather small seat changes in the US House, but is inferring that because abortion appears to have mattered to some voters across the country that it resulted in the pattern we see at the aggregate level. There is no way we can conclude from the results shown here that this caused the seat change pattern we observed or are even correlated with it. In particular, a seat change model would need to consider the massive gerrymandering going on across the country that must be a large factor in understanding Congressional seat change this cycle. Almost certainly Florida, New York, Illinois, etc changes in redistricting were major contributors to what happened in Congress. Here’s an interesting take: https://fivethirtyeight.com/features/redistricting-house-2022/). But regardless, there’s no way we can connect the public opinion data in this paper to what happened to congress without a serious analysis of redistricting, which is a major exogenous shock to the system.

We thank R1 for their comments about redistricting. We have tried to clarify in the text that our paper is not about modeling seat swings, nor about explaining aggregate vote shares across congressional districts. Rather we are simply using the pre-election punditry to help motivate our analysis of our individual-level survey data from the 2020 and 2022 elections. We also added a paragraph on this point in our Discussion. 

The paper is about individual level voting behavior in the 2022 midterm elections, we need to make revisions to the text to make this clear (MIKE). Also point out to the reviewer & editor that redistricting changes would likely have favored the Republicans and are factored into the pundit estimates.

We appreciate R1’s reference to the 538 blog about the post-2020 redistricting, and we note that there are plenty of other reputable blogs that argue that Republicans would benefit from their gerrymandering efforts (examples are the Brennan Center [https://www.brennancenter.org/our-work/analysis-opinion/after-redistricting-heres-how-each-party-could-win-house], the Guardian [https://www.theguardian.com/us-news/ng-interactive/2021/nov/12/gerrymander-redistricting-map-republicans-democrats-visual] and the New York Times [https://www.nytimes.com/2022/09/30/upshot/midterms-gerrymandering-republicans.html]. Our reading of the pundits is that they mostly argue that the Republicans gave themselves an edge going into the 2022 midterms because of redistricting, but as the New York Times points out it was not insurmountable. 

We also would like to note that any partisan advantage that the pundits believed that redistricting would provide before the 2022 midterms would be factored into the pundit estimates for which party would do best in the 2022 midterm election. In this sense, gerrymandering is taken into account in our discussion about how the pundits viewed the 2022 midterm election. 

But to reiterate, our analysis is trying to explain individual-level voting decisions in the 2022 midterm elections, what factors drove voters to vote Republican or Democratic, and how the relative contributions of those factors to voting decisions may have changed between 2020 and 2022. We are not studying vote shares nor seat shifts between elections, nor how votes translate into seats. Those are important questions about contemporary American politics, but they are outside of the scope of the research reported in our paper. 

Another problem is that the authors use the 3 point party id scale, throwing leaners in with true independents. Yet, we know from an abundance of literature that leaners appear to be actually stronger partisans then weak partisans. More weak partisans are likely to defect than leaners. Throwing leaners into the mix of independents is really a problem because in truth many “independents” are really partisans. Thus, the authors have not isolated the appropriate group of voters. They’ve created a large box of voters, most of whom are actually partisans.

We thank Reviewer 1 for noting this important point regarding how we classify voters’ party identification. To address this point, we have re-run our analysis following the convention suggested by the reviewer to show it does not change the topline results of the paper. In this re-analysis,, Republican and Democratic leaners are grouped with their respective parties. For this exercise, only voters who indicate they are independents are classified as such. We show that these changes have only a marginal effect on the topline results.

First, in SI Fig F.2, the share of Democrats and Republicans voting for their respective parties does not meaningfully change from Fig 2. There is, however, a large shift in how the Independents vote — The margin moves from 46-53 in favor of Republicans to 40-59. That is, when excluding leaners, Independents appear significantly more Republican. This means that under this recoding, Independents who are the most persuadable voters strongly preferred the Republican candidate for Congress. 

Second, in SI Fig F.3, we decompose congressional vote preferences by economic and personal financial perceptions. In both cases, the results map to Fig 3 in the main text, with minimal differences. 

Third, in SI Fig F.4, we show that Independent voters are slightly more likely to vote for a Republican in Congress across issues, but this is not surprising given that the voters classified as independent are more Republican leaning than our initial coding. That said, despite small changes in the magnitude of support for Democrats by issue, the direction of how independents voted conditional on the perception of an issue being important is unchanged. 

Next, we replicate the results for Independents probability for voting Republican in SI Fig F.5. Along this dimension, three economic issues are statistically significant – Economic Inequality, the Economy, and Inflation. Stating that abortion is an important issue remains statistically significant and at a slightly larger magnitude (17.5%) compared to the original figure (12.5%). 

Finally, we replicate our main counterfactual in SI Fig F.6. The results for Abortion and Violent Crime are similar to the original text. There is more of a gap between 2020 and 2022 Independents on Immigration, but the Confidence Intervals overlap and the difference is not statistically significant at traditional levels. 

Taken together, although the sample of Independents excluding leaners is 10 percentage points more Republican, the main thrust of the results holds, including the counterfactual in SI Fig F.6. 

In addition, why would abortion shape the federal races given that this was thrown back to the states. I think the authors are probably right abortion did matter in key races –maybe Arizona (for GOV), maybe Michigan (for GOV) (almost for sure with it also on the ballot), maybe Pennsylvania (for SEN) but these were at the state level. I think making this important at the congressional level where incumbency reelection is so high already and information is so low is a hard sell in the face of other issues –election deniers for example, which has been another popular spin on issue importance. I notice the authors did not tap this dimension or the broader threats to democracy on their survey, which seemed to have been a popular issue theme for both Democrats and Republicans.

We appreciate Reviewer 1’s comment. While we would like to include many other issues in our analysis, we are limited by what was included in our survey. Our survey did not include issues like election-denialism and threats to democracy. We discuss this in the discussion section, along with the other limitations suggested by Reviewer 4 (see below). We point out in particular that while it would be interesting to include other issues like these in our analysis, as they are unlikely to be strongly correlated with abortion (which is the central issue variable in our analysis) their omission is unlikely to produce bias in our results. We also note that recent unpublished work (which we cite in the Discussion section) found that election-denialism had only a very slight effect in the 2022 midterm; Malzahn and Hall (2023) summarize their results in the abstract of their working paper by saying that the effect of election-denialism in the 2022 election “is small enough to suggest that only a relatively small group of voters changed their vote in response to having an election-denying candidate on the ballot.” 

There is a large literature on the most important issue question. The authors need to engage that literature especially as it relates to vote choice.

We have added a discussion of the issue importance and vote choice literature and how our paper connects to those past studies.

I found it quite surprising that foreign policy was important in the 2022 election and led to greater support for Republicans. I’m just not sure what to make of that.

We agree that this is an interesting result. Without additional data it is difficult for us to interpret what voters meant when they indicated that foreign policy was important for them – perhaps the war in Ukraine, relations with China, or other dimensions of foreign policy might have been important to voters who then supported Republican congressional candidates. 

Reviewer 2

Reviewer #2: The paper provides a detailed analysis of the 2022 midterm elections, highlighting the discrepancy between the traditional models of midterm elections and the actual election results. The paper suggests that the issue-based model is most consistent with the data gathered from a nationally representative survey of U.S. registered voters. The study further reveals that the Supreme Court’s decision in the Dobbs v. Jackson Women’s Health Organization case played a crucial role in raising the salience of abortion. The conclusion is well-supported by the survey data and provides a plausible explanation for the election results. The paper could have benefited from a more detailed discussion of the limitations of the study, such as the scope of the survey and the potential biases in the sampling process. However, overall, the conclusion is well-argued and provides a comprehensive analysis of the factors influencing the 2022 midterm elections.

Here are some comments on the methodology:

Sampling bias: The survey was conducted online and relied on opt-in survey subject panels. This method of sampling may not represent the entire population, as not everyone has internet access or chooses to participate in surveys.

Non-response bias: Even though the authors attempted to reduce the effects of non-response bias by offering compensation and weighting by various demographic factors, it is still possible that those who choose to respond to surveys are not representative of the population.

Self-reported bias: The authors used self-reported data to measure economic and national economy evaluations, which may be subject to bias.

Timing bias: Although the authors attempted to reduce the effects of media narratives by collecting data in the days immediately following the election, it is still possible that media coverage and other external factors could have influenced respondents' answers.

Limitations of weighting: The authors used weighting to adjust for various demographic factors, but this method may not fully address the potential biases in the sample.

Limited generalizability: The survey was conducted in November 2022, immediately following the 2022 midterm election, and may not be generalizable to other time periods or elections

We extend our gratitude to R2 for their valuable feedback. We have taken their comments into serious consideration and addressed them, along with other limitations of our study, in the Data and Methodology section of the paper. Specifically, we have provided detailed explanations concerning potential sampling bias, selection bias, and non-response issues. Moreover, we have outlined the measures we have implemented to mitigate these concerns while remaining transparent about any issues that may persist. Furthermore, in interpreting our results, we exercise caution and precision, emphasizing the unique circumstances surrounding the 2022 Congressional midterm elections. This approach ensures that our conclusions are grounded in the context of that particular time period. We acknowledge that the dynamics we have identified are subject to change over time, and therefore, we conclude by highlighting the necessity for further research to assess these dynamics over a more extended period.

Reviewer 3

Major comments:

- While I am convinced that issues are important correlates to voter preference, I would be cautious to say that issues explain the election outcome. When the authors said strong statements, such as “Persuadable voters decided the 2022 midterm” I took pause becayse the data does not necessarily agree with this statement. Instead, even if independent's choices changed, we do not know the number of independents who were inspired to vote because of the abortion debate. In other words, the Dobbs decision's impact could have been a stronger motivation for Democrats to vote, and the change in the votes from the Independents who actually voted could have been minimal (I would be more convinced had voting been a requirement, as it is in many countries, in which case motivation would not have been as big a factor). I do not think this is a fundamental flaw, but rather something that is worth mentioning, and therefore I would be cautious to use as strong a title as the current one.

Another critique is the idea that abortion was the deciding issue, given a similar effect size was seen for violent crime. How can we say that abortion drove a change but violent crimes did not cancel out this effect, given it too was a larger factor in 2022 than 2020? If both factors were equally important, then they would cancel each other out, and the reason for the Democrats' win would be presumably higher motivation for Democrats to vote than Republicans. To address this fundamental problem, I suggest the authors create a simulation to show a counter-factual. Cf.

Daniel Hickey, Matheus Schmitz, Daniel Fessler, Paul Smaldino, Goran Muric, Keith Burghardt. (2023). No Love Among Haters: Negative Interactions Reduce Hate Community Engagement. arXiv preprint arXiv: 2303.13641

The counter-factual will utilize the model the authors already created; the actual effort to create this counter-factual is relatively minor.

1. In the present paper, the authors could show the simulated % of voters voting Democrat in 2022. This is the logic model applied to each datapoint. One datapoint with features X1 will have a probability p1 of voting Democrat. Another datapoint with feature X2 will have probability p2 of voting democrat. The expected number to vote Democrat will be p1+p2. After applying the model to each datapoint and summing all the probabilities together, you will have the expected percent of voters who will vote Democrat: (p1+p2+p3+...)/(number of people polled). This percentage can be compared with the actual data - e.g., you will predict 55% of voters will vote Democrat, but the data shows a 54% percent. This will indicate that the model correctly estimates who votes for whom.

2. Next, they can compare this model to one in which all parameters are the same but the coefficient for abortion = 0 (i.e., abortion were forced to have no effect on voting preferences), and similarly a simulation where all coefficients are the same except violent crime is set to 0. This will adjust each datapoint's probability to vote Democrat. Taking the sum of probabilities will create the appropriate counter-factual condition. We should see:

• <50% of voters voting Democrat if abortion coefficient is set to zero

• >50% of voters voting Democrat if the violent crime coefficient is set to zero (and the absolute change compared to status quo is minimal)

The reason I argue this is an important contribution is again because violent crime and abortion are both major issues. If the impact of violent crime was small in an absolute sense then removing that coefficient would not strongly impact the percent of voters voting Democrat. Alternatively, if the abortion importance coefficient were set to 0, we might expect the % of voters voting Democrat to change dramatically. In contrast to the covariates extracted from your model, this simulation accounts for the relative number of people who are Democrats, Independents, and Republicans, as well as the number who consider each issue important.

We thank the reviewer for this response. We ran the counterfactual as outlined. However, we decided to not include the results in the paper as we feel that it does not comment on our thesis, but a parallel matter. The original model correctly predicts that 50.2% of voters will vote for the Democratic candidate. When the coefficient on abortion importance is set to 0, which is the same as saying no one believed abortion to be important, the new result says 46.4% of the electorate will vote for the Democratic candidate. This is a 3.8% swing in favor of the Republicans. If the coefficient on violent crime is set to 0, 56.4% of the electorate would vote for the Democratic candidate—a 6.2% swing in favor of the Democrats. 

However, this difference is the reason we used the counterfactual from 2020 rather than setting the coefficients to zero in our original explanation. We are arguing that compared to previous years, abortion has an outsized effect on vote choice. In both elections violent crime has a significant coefficient, but the coefficient does not change between years. 

A more realistic comparison, in line with the conversation in the paper, would be, rather than setting the coefficients to the values found in the 2020 election rather than 0 as the counterfactual. If that is the case, when looking at the abortion coefficient being set to 2020 levels, the Democrats would only receive 45.2% of the national vote. While if the violent crime coefficients were set to 2020 levels they would receive 55% of the national vote. While this analysis is interesting, it seemed more complicated than the results presented in Figure 8, without much additional information provided, 

Reviewer 4

I would recommend;

• A section on the literature review on strategic voting behavior.

• The text is very dense and can be shortened somewhat. In my opinion, the paper could benefit from highlighting the hypotheses tested (explicitly stated) so that the reader can easily understand the procedure chosen by the authors.

In my opinion, the paper could benefit from presenting and commenting on the results after testing the hypotheses

• Empirical strategy : The model specification is very intransparent. The reader wants to take a look at the estimates and at the figures and tables and needs to understand immediately what it is all about. I suggest that the authors first write down the equation of the model to be estimated and then present the estimation table, which needs to be commented on right after. This will make the empirical strategy more transparent. Then, I would suggest describing the most basic model and later come up with a more complicated equation or just verbally state what other terms are added to this basic model. In general, I do not understand exactly what these estimates mean because of the intransparency presentation of the estimates. Therefore, it is difficult for me to comment on the interpretation of the results later. However, I like the graphical representation of the results.

We thank the Reviewer for these comments. 

Respectfully, most of the literature on strategic voting does not directly apply to our current work. While we do discuss sophisticated voting models like policy balancing in our paper, our focus is on Congressional elections, which typically involve only two major candidates — one Democratic and one Republican. As such, the opportunity for strategic voting by voters is minimal. In a two-candidate election, the concept of strategic voting, which involves voting for a candidate other than one's favorite to improve the chances of defeating a less-preferred candidate, is not applicable. Such an action would essentially mean voting for someone who is the voter's least favorite candidate, which is never an optimal strategy. While the literature on strategic voting may have relevance in the context of primary elections, where more than two candidates compete, it is not within the scope of our current analysis. Therefore, we briefly acknowledge this aspect along with other limitations and explain why we consider it irrelevant for our present study.

Second, we have edited the paper to make the methodology more transparent, including the equations that the Reviewer requested.

MINOR:

1. In the third last line on p.4 : "Organization decision, which held that the the Constitution does not provide a right to" there is an additional "the"

We made this edit. 

3. The regression table D needs a little reformatting because the title and the columns do not match

We have edited the column labels in this table.

---

## [Decision Letter · Decision Letter 1]

25 Oct 2023

Persuadable voters decided the 2022 midterm: Abortion rights and issues-based frameworks for studying election outcomes

PONE-D-23-07439R1

Dear Dr. Ebanks,

We’re pleased to inform you that your manuscript has been judged scientifically suitable for publication and will be formally accepted for publication once it meets all outstanding technical requirements.

Kind regards,

Carlos Henrique Gomes Ferreira, Ph.D.

Academic Editor

PLOS ONE

Additional Editor Comments (optional):

I would like to express my appreciation to the authors for addressing the necessary revisions highlighted by the reviewers. I thank you for your diligent efforts in this regard.

Reviewers' comments:

Reviewer's Responses to Questions

**Comments to the Author**

1. If the authors have adequately addressed your comments raised in a previous round of review and you feel that this manuscript is now acceptable for publication, you may indicate that here to bypass the “Comments to the Author” section, enter your conflict of interest statement in the “Confidential to Editor” section, and submit your "Accept" recommendation.

Reviewer #3: All comments have been addressed

Reviewer #4: All comments have been addressed

2. Is the manuscript technically sound, and do the data support the conclusions?

Reviewer #3: Yes

Reviewer #4: Yes

3. Has the statistical analysis been performed appropriately and rigorously? 

Reviewer #3: Yes

Reviewer #4: Yes

4. Have the authors made all data underlying the findings in their manuscript fully available?

Reviewer #3: Yes

Reviewer #4: Yes

5. Is the manuscript presented in an intelligible fashion and written in standard English?

Reviewer #3: Yes

Reviewer #4: Yes

6. Review Comments to the Author

Reviewer #3: I respect the convincing simulation the authors performed, and I believe that the suggestions made by myself and other reviewers have been addressed.

Reviewer #4: As a reviewer for this paper, I am very pleased with the authors' work in answering each question. The authors demonstrated a deep understanding of the topic and addressed all concerns and comments raised during the initial review process. Their responses are clear, well organised, and supported by appropriate references and evidence.

It is evident that the authors have made significant efforts to refine their work, and their responses to the reviewers' suggestions demonstrate their commitment to a high-quality final manuscript.

Overall, I look forward to seeing the final version of the article. Well done!

7. PLOS authors have the option to publish the peer review history of their article (what does this mean?). If published, this will include your full peer review and any attached files.

Reviewer #3: No

Reviewer #4: No

---

## [Editor Report · Acceptance letter]

3 Nov 2023

PONE-D-23-07439R1 

Persuadable voters decided the 2022 midterm: Abortion rights and issues-based frameworks for studying election outcomes 

Dear Dr. Ebanks:

I'm pleased to inform you that your manuscript has been deemed suitable for publication in PLOS ONE. Congratulations! Your manuscript is now with our production department. 

Kind regards, 

on behalf of

Dr. Carlos Henrique Gomes Ferreira 

Academic Editor

PLOS ONE